Manuscript prepared for Atmos. Chem. Phys.
with version 2014/09/16 7.15 Copernicus papers of the LATEX class copernicus.cls.
Date: 24 November 2016

# OH reactivity at a rural site (Wangdu) in the North China Plain: Contributions from OH reactants and experimental OH budget

Hendrik Fuchs[1], Zhaofeng Tan[2], Keding Lu[2], Birger Bohn[1], Sebastian Broch[1],
Steven S. Brown[3], Huabin Dong[2], Sebastian Gomm[1,a], Rolf Häseler[1],
Lingyan He[4], Andreas Hofzumahaus[1], Frank Holland[1], Xin Li[1,b], Ying Liu[2],
Sihua Lu[2], Kyung-Eun Min[3,5,c], Franz Rohrer[1], Min Shao[2], Baolin Wang[2],
Ming Wang[6], Yusheng Wu[2], Limin Zeng[2], Yinson Zhang[2], Andreas Wahner[1], and
Yuanhang Zhang[2,7]

[1]Institute of Energy and Climate Research, IEK-8: Troposphere, Forschungszentrum Jülich GmbH,
Jülich, Germany
[2]College of Environmental Sciences and Engineering, Peking University, Beijing, China
[3]Chemical Sciences Division, Earth System Research Laboratory, National Oceanic and
Atmospheric Administration, Boulder, CO, USA
[4]Key Laboratory for Urban Habitat Environmental Science and Technology, School of
Environment and Energy, Peking University Shenzhen Graduate School, Shenzhen, China
[5]Cooperative Institute for Research in Environmental Sciences, University of Colorado, Boulder,
CO, USA
[6]School of Environmental Sciences and Engineering, Nanjing University of Information Science
and Technology, Nanjing, China
[7]CAS Center for Excellence in Regional Atmospheric Environment, Chinese Academy of Science,
China
[a]now at: d-fine GmbH, Opernplatz 2, 60313 Frankfurt, Germany
[b]now at: College of Environmental Sciences and Engineering, Peking University, Beijing, China
[c]now at: School of Environmental Science and Engineering, Gwangju Institute of Science and
Technology, Gwangju, Korea

*Correspondence to:* H. Fuchs (h.fuchs@fz-juelich.de), Y. Zhang (yhzhang@pku.edu.cn)

**Abstract.** In 2014, a large, comprehensive field campaign was conducted in the densely populated
North China Plain. The measurement site was located in a botanic garden close to the smaller town
Wangdu without major industry, but influenced by regional transportation of air pollution. The loss
rate coefficient of atmospheric hydroxyl radicals (OH) was quantified by direct measurements of
the OH reactivity. Values ranged between 10 and $20\,\mathrm{s}^{-1}$ for most of the daytime. Highest values
were reached in the late night with maximum values around $40\,\mathrm{s}^{-1}$. OH reactants mainly originated
from anthropogenic activities as indicated (1) by a good correlation between measured OH reactiv-
ity and carbon monoxide (linear correlation coefficient $R^2 = 0.33$), and (2) by a high contribution
of nitrogen oxide species to the OH reactivity (up to 30 % in the morning). Total OH reactivity was
measured by a laser flash photolysis - laser induced fluorescence instrument (LP-LIF). Measured
values can be well explained by measured trace gas concentrations including organic compounds,
oxygenated organic compounds, CO and nitrogen oxides. Significant, unexplained OH reactivity

was only observed during nights, when biomass burning of agricultural waste occurred on surrounding fields. OH reactivity measurements also allow investigating the chemical OH budget. During this campaign, the OH destruction rate calculated from measured OH reactivity and measured OH concentration was balanced by the sum of OH production from ozone and nitrous acid photolysis and OH regeneration from hydroperoxy radicals within the uncertainty of measurements. However, also a tendency for higher OH destruction compared to OH production at lower concentrations of nitric oxide is observed consistent with previous findings in field campaigns in China.

## 1 Introduction

Hydroxyl radicals (OH) are the most important oxidizing agent for inorganic and organic pollutants in the atmosphere (Ehhalt, 1999). A large number of field campaigns have been conducted in the past to improve our understanding of radical chemistry in the atmosphere at various locations all over the world (Rohrer et al., 2014). However, only few have taken place in China, where air pollution is still a severe problem (Lu et al., 2010). Measurements during field campaigns in the Pearl-River-Delta (PRD) and at a suburban location south of Beijing (Yufa) revealed a lack of understanding of radical chemistry by state-of-the-art chemical models pointing to unknown OH radical sources (Hofzumahaus et al., 2009; Lu et al., 2012, 2013). Similar results were found at other locations, which were mainly dominated by biogenic emissions (Rohrer et al., 2014).

In summer 2014, the effort to improve our knowledge of radical chemistry in Chinese megacity areas was continued by a comprehensive field campaign at a location close to the city Wangdu in the North China Plain south-west of Beijing (Tan et al., 2016). A large set of instruments was deployed to detect radicals (OH, $HO_2$, $RO_2$), reactive trace gases (e.g., CO, $NO_x$, volatile organic compounds (VOC)) and aerosol properties. Compared to our previous field campaigns in China 2006 (Hofzumahaus et al., 2009), the quality and number of measurements have been improved. A large number of instruments measured a variety of different trace gases, part of which were simultaneously detected by several instruments. Specifically, measurements of organic oxygenated compounds such as formaldehyde and acetaldehyde were achieved, which was not the case in previous campaigns. Radical measurements were improved by performing additional tests of potential interferences in the detection of OH and a modified the detection scheme for $HO_2$ that avoids interference from $RO_2$ was applied (Fuchs et al., 2011). Time series of radical measurements and a comparison with results from a chemical box model calculation are discussed in our accompanying paper by Tan et al. (2016).

OH reactivity ($k_{OH}$) is the pseudo-first order loss rate coefficient of OH radicals and represents the inverse chemical lifetime of OH.

$$k_{OH} = \sum_i k_{OH+Xi}[X_i] \tag{1}$$

$X_i$ represents any OH reactant. Because of the large number of OH reactant in the atmosphere, it is of high value for the interpretation of radical chemistry to compare the direct measurement of $k_{OH}$ with reactivities calculated from measured atmospheric OH reactant concentrations. The difference measured and calculated reactivity is often referred to as missing reactivity.

Depending on the instrumentation that were available in field campaigns in the past, up to more than 70 % of the measured reactivity was found to remain unexplained in different types of environments (e.g., cities, forests) (Yang et al., 2016). For our previous field campaigns in China, the measured OH reactivity was two times larger than the calculated $k_{OH}$. The discrepancy could be quantitatively explained by the reactivity from oxygenated VOCs (OVOC), which were not mea-

sured, but estimated by a chemical model (Lou et al., 2010; Lu et al., 2013). In this campaign, the number of measured species was extended and included important atmospheric OVOCs, for example formaldehyde, acetaldehyde, isoprene oxidation products (methyl-vinyl ketone and methacrolein), and glyoxal.

Measurements of OH reactivity and OH concentrations can be combined to calculate the loss rate of OH radicals. This can then be compared to the sum of OH production rates from ozone and nitrous acid photolysis, reaction of hydroperoxy radicals with nitric oxide and ozone as well as ozonolysis reactions of alkenes. All quantities that are required to do this calculation were measured in this campaign. This allows for a model-independent analysis of the chemical OH budget. This approach was successfully applied to quantify unaccounted OH production in our field campaigns in China in 2006 (Hofzumahaus et al., 2009).

In the following, we describe the technique for OH reactivity measurements applied in the campaign in Wangdu, discuss the time series of measurements, compare OH reactivity measurements with calculations from single reactant measurements and analyze the OH budget.

## 2 Experimental

The instruments, their setup at the field site and the measurement conditions are described in Tan et al. (2016). Therefore, only a brief description is given here.

### 2.1 Measurement site

Measurements took place inside a botanic garden close to the small town Wangdu in China between 7 June and 8 July 2014. Wangdu is located in the densely populated North-China Plain, but does not have major industry itself. Major cities are located mainly in the sector from north-east to south-west from Wangdu, whereas there is a mountainous area with less industry north-west of Wangdu. The closest large city is Baoding 35 km north-east of Wangdu. The measurement site had a distance of 2 km from a road with only local traffic. The botanic garden was surrounded by agricultural fields. Trace gases from local biogenic emissions of trees, bushes and from farming can be expected.

The site was chosen, because it was not directly influenced by strong close-by anthropogenic emissions or the direct outflow of a big city. However, it was expected to observe regionally transported pollution in the North China Plain.

Instruments were housed in seven shipping containers, which were partly stacked up, so that inlets of instruments were at a height of 7 m above the ground.

### 2.2 Instrumentation

A large number of instruments characterized meteorological conditions, trace gas concentrations and aerosol properties. The measurements used for the OH reactivity analysis are listed in Table 1.

OH and $HO_2$ radical concentrations were measured by a newly built instrument applying laser-induced fluorescence technique (PKU-LIF) (Tan et al., 2016). This instrument detects OH fluorescence by time-delayed single photon counting after excitation by short laser pulses at 308 nm in a low-pressure cell (Holland et al., 2003; Fuchs et al., 2011). $HO_2$ radicals are detected as the sum of OH and $HO_2$ ($=HO_x$) after chemical conversion to OH in the reaction with nitric oxide (NO). In order to avoid significant simultaneous conversion of organic peroxy radicals ($RO_2$) (Fuchs et al., 2011), the amount of NO was adjusted to yield an $HO_2$ conversion efficiency of only 6 %. The instrument sensitivity was calibrated every 3 to 4 days by a custom-built calibration source described in detail in Fuchs et al. (2011).

A commercial cavity ring-down instrument (Picarro model G2401) monitored CO, $CH_4$ and $H_2O$ concentrations. Concentration measurements of ozone by two commercial UV absorption instruments (Environment S.A. model 41M; Thermo Electron model 49i) agreed well within their accuracies during the campaign. Nitrogen oxides (NO and $NO_2$) were also detected by several instruments applying chemiluminescence technique (Thermo Electron model 42i NO-$NO_2$-$NO_x$ analyzer and Eco Physics model TR 780) that were equipped with a photolytic converter. Daily calibrations were performed using a certified gas standard. The field measurements differed on average by 20 %. Measurements of the Thermo Electron instruments appeared to be more precise and are used here (see Tan et al. (2016) for details). Because the reason for the disagreement could not be identified, the 20 % difference adds to the uncertainty of NO measurements here.

Nitrous acid (HONO) concentrations were simultaneously measured by several instruments applying different measurement techniques (Tan et al., 2016). Custom-built instruments from FZJ (Li et al., 2014) and from PKU (Liu et al., 2016) utilized long-path absorption photometry (LOPAP). In addition, three custom-built instruments applied cavity enhanced absorption spectroscopy (CEAS) for the detection of HONO. They were operated by the US National Oceanic and Atmospheric Administration (NOAA) (Min et al., 2016), by the Anhui Institute of Optics and Fine Mechanics (AIOFM), and by the University of Shanghai for Science and Technology (USST). A gas and aerosol collector (GAC), which is based on the wet denuder/ion chromatography technique, could also detect HONO (Dong et al., 2012). Only measurements by the two LOPAP instruments and the CEAS by NOAA resulted in good data coverage. The agreement between these instruments was diverse. Differences were often less than 30 %, but could be as high as a factor of two for certain periods (several hours). The reason for the disagreement during these times is not clear. For the purpose of the analysis here, measurements by the LOPAP instrument from Forschungszentrum Jülich are used (Li et al., 2014) because this instrument showed best data coverage and the lowest detection limit. This instrument was calibrated by using a liquid standard as described in Li et al. (2014) every ten days. The choice of the HONO data set has a rather small impact on the calculated OH reactivity, as well as on the calculated total OH production rate which was dominated by OH recycling from $HO_2$ during daytime (see below).

For the analysis of the OH reactivity, measurements of organic trace gases are essential. 59 organic species ($C_2$-$C_{11}$ alkanes, $C_2$-$C_6$ alkenes, $C_6$-$C_{10}$ aromatics, and isoprene) were detected by a custom-built gas-chromatography system equipped with a flame ionization detector (FID) (Wang et al., 2014). Full calibrations using certified gas standards (Air Environmental Inc., Spectra Gases Inc.) were done before and after the campaign. Drifts of the sensitivity during the campaign were accounted for by measuring the instrument sensitivity for bromochloromethane, 1,4-difluorobenzene, chlorobenzene, and 1-bromo-3-fluorobenzene every second day. Formaldehyde (HCHO) was detected by a commercial Hantzsch monitor (Aerolaser model AL4021) and glyoxal (CHOCHO) by a custom-built cavity enhanced spectrometer (Min et al., 2016). In addition, acetaldehyde and the sum of methyl vinyl ketone (MVK) and methacrolein (MACR) were measured by a commercial proton transfer reaction - mass spectrometry system (PTR-MS, Ionicon). Some of the species or family species were simultaneously detected by the GC system and the PTR-MS (isoprene, benzene, toluene, styrene, $C_8$-aromatics, $C_9$-aromatics). Measurements during daytime well agreed within 30 to 50 % (Tan et al., 2016). Calibration of the PTR-MS instrument was done every day using a certified gas standard (Air Environmental Inc.).

Photolysis frequencies were calculated from the spectral actinic photon flux density measured by a spectrometer that was calibrated against absolute irradiance standards (Bohn et al., 2008).

## 2.3 OH reactivity measurements

The OH reactivity instrument measures directly pseudo first-order loss rate coefficients (Eq. 1) of OH in the ambient air. The measurement is based on artificial OH generation by pulsed laser-flash photolysis (LP) of ozone in ambient air combined with the detection of the temporal OH decay by laser induced fluorescence (LIF). The method was initially developed for field application by Sadanaga et al. (2004) and is applied today by several other groups (Lou et al., 2010; Parker et al., 2011; Stone et al., 2016). The instrument deployed in this campaign is similar to the instrument described in Lou et al. (2010), which was used for measurements in our two field campaigns in 2006 in China. Since then, a second instrument was built specifically for the deployment on a Zeppelin NT airship (Li et al., 2014), but can also be operated at ground. This instrument was deployed. Figure 1 gives a schematic representation of the instrument without the pump (Edwards model XDS35i) needed for the operation of the low-pressure LIF cell and without the laser that provides the 308 nm radiation for the excitation of OH. The 308 nm radiation is delivered by the dye laser system that is also used in the instrument for the OH HO$_2$, and RO$_2$ concentration measurements described in Tan et al. (2016). This laser has three output fibers to provide laser light, one of which is used for the OH reactivity instrument.

The $k_{OH}$ instrument is mounted in a 19" rack that was placed inside one of the upper shipping containers at the field site. The inlet line (outer diameter 10 mm, length approximately 6 m) was made of stainless steel that had a SilcoNert 2000 coating. Such sampling line has been used for OH reac-

tivity measurements in the Jülich atmosphere simulation chamber SAPHIR for many years without notable effects on measurements. Approximately $20\,\mathrm{Liter/min}$ of ambient air is sampled through a flow-tube made of anodized aluminium (length: $60\,\mathrm{cm}$, inner diameter: $4\,\mathrm{cm}$). Downstream of the flow tube, the flow rate is measured by a flow-meter and controlled by a blower.

The pressure inside the flow tube is $1\,\mathrm{atm}$ and the temperature was the same as in the field container (between 22 and $30\,^\circ\mathrm{C}$). Ambient temperature was higher with up to $38\,^\circ\mathrm{C}$ for some periods during the campaign. Differences in temperature and pressure potentially effect the measured reactivity due to changes of the reactant concentrations and of reaction rate constants (Lou et al., 2010). Measured reactivities were corrected for changes in the reactant concentration calculated from measured ambient and flow-tube temperature and pressure values (corrections were less than $2\,\%$). Sensitivity studies taking either ambient temperature or flow-tube temperature for the calculation of OH reactivity from measured OH reactant concentrations (see below) indicate that the effect of temperature differences on reaction rate constants resulted in changes in the OH reactivity of typical less than $1\,\%$ (maximum values $4\,\%$) for conditions of this campaign.

High OH concentrations on the order of $10^9\,\mathrm{cm}^{-3}$ are produced by flash photolysis of $O_3$ at $266\,\mathrm{nm}$ with subsequent reaction of $O^1D$ with water vapor. The $266\,\mathrm{nm}$ laser pulses (pulse energy 20 to $28\,\mathrm{mJ}$, repetition rate $1\,\mathrm{Hz}$, pulse duration less than $10\,\mathrm{ns}$) are provided by a compact, frequency quadrupled Nd:YAG laser (Quantel model Ultra 100). The laser is mounted on one side of an optical rail, on which the flow tube is mounted on the opposite side. The laser beam is widened by an optical telescope to a diameter of $3\,\mathrm{cm}$ and guided to the flow tube by two turning mirrors.

Water vapor, temperature and pressure in the flow tube are continuously monitored. Normally, ozone and water vapor concentrations in the sampled ambient air are sufficiently high in order to produce high OH concentrations. However, ozone can be depleted during night due to its reaction with nitric oxide and by deposition processes. Therefore, a small flow of synthetic air ($0.2\,\mathrm{Liter/min}$) that has passed an ozonizer (glass tube of fussed silica with a mercury lamp providing $185\,\mathrm{nm}$ radiation) can be added, in order to increase ozone mixing ratios in the flow-tube by 40-50 ppbv. The injection is controlled by a solenoid valve which is automatically opened, if the ozone mixing ratio in ambient air drops below $30\,\mathrm{ppbv}$.

At a distance of $48\,\mathrm{cm}$ from the inlet of the flow-tube, $1\,\mathrm{Liter/min}$ of the total flow is sampled from the center of the flow tube through a conical nozzle into the OH detection cell. The design of the OH fluorescence cell is the same as used for OH concentration measurements (Tan et al., 2016).

In the cell, OH is excited by $308\,\mathrm{nm}$ radiation from a tunable frequency-doubled dye laser, which is operated at a pulse repetition rate of $8.5\,\mathrm{kHz}$. The OH fluorescence is detected by gated photon counting and accumulated in time bins of $0.6\,\mathrm{ms}$. This way, the chemical decay of OH in the flow tube is recorded for $1\,\mathrm{s}$ after the photolysis laser pulse. For photon detection, a gated multichannel photomultiplier (Photek, PM325) is used in combination with a multichannel counting card (Sigma Space, AMCS).

In order to achieve sufficiently precise reactivity measurements, 60 decay curves are taken for one reactivity measurement resulting in an amplitude of 50 to 100 counts of the decay curve. Because of the scanning of the laser over the absorption line of OH in order to track slow drifts in the wavelength of laser, the amplitude of the decay curve changes periodically. Therefore, ten OH decay curves are summed up to equalize the amplitude. Six of the summed curves are then averaged to determine realistic error estimates needed for the fit procedure. A weighted single-exponential fit (Levenberg-Marquardt minimization) is then applied to derive the OH reactivity (Eq. 1). Approximately the first 30 to $50\,\mathrm{ms}$ of the decay curve are not included in the fit, because these points deviate from the single-exponential behavior that is observed at later times. The fit is started, if the count rate has decreased to the $90\,\%$ level of the maximum count rate. The likely reason for an inhomogeneous initial OH distribution is that the spatial OH distribution is not perfectly homogeneous near the inlet nozzle of the OH detection cell right after the laser pulse due to inhomogeneities in the laser power across the laser beam.

Diffusion to the wall of the flow tube, where OH is lost by wall reactions, causes loss of OH even in the absence of OH reactant. This zero loss rate is regularly measured in humidified air (purity $99.999\,\%$). Typical zero loss rates measured in laboratory characterization measurements are around $3\,\mathrm{s}^{-1}$ for this instrument. A slightly higher value of $3.8\,\mathrm{s}^{-1}$ was derived in measurements sampling synthetic air from a gas cylinder during the campaign. Analysis of the synthetic air in this gas cylinder by gas-chromatography yielded contaminations with an OH reactivity of $0.7\,\mathrm{s}^{-1}$. Therefore, an instrumental zero decay value of $3.1\,\mathrm{s}^{-1}$ was subtracted from ambient OH reactivity measurements consistent with previous values for this instrument. The reactivity measured in the synthetic air is considered as a potential systematic error of the OH reactivity measurements in this campaign. The accuracy of our LP-LIF technique has been tested with CO and $CH_4$ mixtures in synthetic air. Measured $k_{OH}$ agreed better than $10\,\%$ with the expected, calculated OH reactivity for values up to $60\,\mathrm{s}^{-1}$ in agreement with previous studies by Lou et al. (2010). At higher $k_{OH}$ values, the initial non-exponential part of the OH decay curve starts to influence the quality of the fitted OH decay curve, but such high $k_{OH}$ values were not encountered in the campaign at Wangdu (Fig. 2).

Potential interferences that could be present in the OH concentration detection would not affect the measured OH reactivity, because OH that would be artificially produced inside the measurement cell would only increase the background signal, but not the decay time as long as it does not change on the time scale of the OH decay measurement ($1\,\mathrm{s}$). In any case, however, effects are expected to be negligible due to the high OH concentration inside the flow tube that are much higher compared to ambient OH concentrations, for which interferences have been recognized. This holds for the known interference from ozone photolysis by the $308\,\mathrm{nm}$ laser radiation, but also for other potential interferences that have been reported for OH concentration measurements (Mao et al., 2012; Novelli et al., 2014) and which could not be fully excluded for this campaign (Tan et al., 2016).

If ambient NO concentrations are high enough to lead to a significant regeneration of OH from secondarily formed $HO_2$, the shape of the decay curve changes to a bi-exponential behavior. This can be derived from reaction kinetics. The faster decay time represents approximately the OH reactivity for certain chemical conditions. As shown in Lou et al. (2010) no significant effects are expected for NO mixing ratios of up to 20 ppbv for realistic OH reactant mixtures in our instrument. During the

campaign in Wangdu, NO mixing ratios were generally well below 20 ppbv. No bi-exponential behavior was observed that would have been seen in the residuum of the fit. NO mixing ratios exceeded 20 ppbv only for some short periods mainly during nighttime on three days, but measurements still appeared as single exponential decays in these cases.

## 3   Results and Discussion

### 3.1   Time series of OH reactivity

Measured OH reactivity values ranged between 10 and 20 $s^{-1}$ during this campaign for most of the time (Fig. 2). In general, values were lower during daytime (median value 12.4 $s^{-1}$) than at night (median value 15.4 $s^{-1}$). During the first two weeks, midday OH reactivity increased from 10 $s^{-1}$ on 8 June to values higher than 20 $s^{-1}$ between 15 and 19 June. After 19 June, OH reactivity was

generally lower and more uniform till the end of the campaign.

    Maximum values were observed during nighttime and early morning hours, when OH reactivities show spikes with values of up to 60 $s^{-1}$ for short periods of less than one hour. The high reactivity values were probably caused by emissions into the shallow nocturnal boundary layer. The short duration indicates that nearby local sources were responsible for these events. This happened more

frequently during the first part of the campaign and only few spikes were observed after 19 June.

    The overall changes in OH reactivity values from day to day were likely dominated by anthropogenic activities during this campaign. The measured OH reactivities show an increasing trend with CO, which cannot be explained by the reactivity of CO alone (Fig. 3). Therefore, other reactants that were co-emitted with CO for example in combustion processes most likely contributed to the

increase in reactivity. The correlation still holds, if only reactivity from measured OH reactants other than CO, $NO_x$ and isoprene is taken into account. This further supports that also OH reactivity from organic compounds is co-emitted with CO.

    Back-trajectories were calculated for this campaign using the NOAA (Nation Oceanic and Atmospheric Administration) HYSPLIT (Hybrid Single Particle Lagrangian Integrated Trajectory Model)

model (Stein et al., 2015), in order to test, if measured OH reactivities are correlated with the origin of advected air masses. 24-hour back-trajectories were calculated for air masses at the measurement site for each hour. During most days, back-trajectories were very similar Therefore, trajectories between 10:00 and 19:00 were averaged (Fig. 4). The majority of back-trajectories are pointing to locations south of Wangdu and less often to locations east or north of the measurement site. Moun-

tains that are west and north of the measurements site appear as barriers for air masses. Only on three days (08, 27, 28 June) back-trajectories indicate that air masses originated from locations in the mountains. Lowest $k_{OH}$ values ($< 10\,\mathrm{s}^{-1}$) were observed in these cases due to less emissions from industry and from other anthropogenic activities. In contrast, there is dense population east and south of the measurements site. This likely explains why OH reactivity values were highest, if air masses were coming from this area. Also the relation between $k_{OH}$ and CO is consistent with the assumption that OH reactivity was dominated by anthropogenic activities in this case.

The increase in OH reactivity during the first two weeks could be related to a change of the origin of air masses from the north (08 June) to the east (13 June) and finally to the south (15 June). However, no obvious difference between back-trajectories is seen before and after 20 June, so that back-trajectories are not sufficient to explain, why measured OH reactivity would be higher and more spiky before 20 June.

The more likely reason for differences in OH reactivity is emissions connected with harvesting of crop and combustion of straw and crop residuals on nearby agricultural fields in the first two weeks of June. On 13 June, for example, crop was harvested on the field directly next to the measurement place. Indicators for biomass burning activities were visually observed fires, reduced visibility, and an increase in measured particle number concentrations. Typical daytime maximum PM2.5 concentrations ranged between 30 and $90\,\mu\mathrm{g/m^3}$ but were as high as $300\,\mu\mathrm{g/m^3}$ on one day due to the local biomass burning (Fig. 2). No clear connection between OH reactivity and aerosol number concentration was observed. Although a sharp drop in PM2.5 was observed on 19 June when also OH reactivity dropped, PM2.5 increased again to higher values till the end of the campaign. Elevated concentrations of acetonitrile (a marker for biomass combustion) were measured between 12 and 19 June (Tan et al., 2016).

### 3.2 Contributions of OH reactants to the OH reactivity and missing reactivity

OH reactivity measurements are of particular value in order to test if all important OH reactants were detected. Volatile organic compounds (VOCs) and inorganic compounds such as nitrogen oxides ($NO_x$=NO+$NO_2$) and carbon monoxide (CO) are typically major contributors to the total OH reactivity. However, the number of OH reactants, specifically of organic compounds is very large, so that a complete measurement is not expected. Therefore, comparison of direct $k_{OH}$ measurements with calculations from measured reactants can reveal to which extend unmeasured reactive compounds contributed to total OH reactivity. This presents a gap in the constraints of model calculations used to test our knowledge of radical chemistry (Tan et al., 2016). In addition, VOCs and $NO_x$ are key species for understanding ozone and particle formation, so that an incomplete knowledge of OH reactivity would lead to a systematic underprediction of ozone production by chemical models (e.g., Whalley et al. (2016); Griffith et al. (2016)).

The full time series of the calculated $k_{\mathrm{OH}}$ is plotted together with the measured total $k_{\mathrm{OH}}$ in Fig. 2. The calculated reactivities were determined from measured CO, $CH_4$, $C_2$ to $C_{11}$ alkanes, $C_2$ to $C_6$ alkenes, $C_6$ to $C_{10}$ aromatics, formaldehyde, glyoxal, acetaldehyde, MVK, MACR, NO, $NO_2$, $SO_2$ (Table 1). Reaction rate constants were taken from IUPAC recommendations (Atkinson et al., 2013) or structure-activity relationship (SAR) as stated in the Master Chemical Model (http://mcm.leeds.ac.uk/MCM/).

During each of the two parts of the campaign (before and after 19 June), diurnal profiles of observations appear to be similar. Therefore measured $k_{\mathrm{OH}}$ and calculated reactivity from major contributors are shown as median diurnal profiles with percentiles for each period in Fig. 5. Median diurnal profiles of all measured contributions are summed up and compared to measured $k_{\mathrm{OH}}$ in Fig. 6. Ambient temperature was used for the calculation of reaction rate constants, but the differences between ambient temperature and the actual temperature in the instrument does not change any of the results shown here.

The most important OH reactants were CO (on average 20 to 25 % of the total OH reactivity), nitrogen oxides (on average 12 to 22 % of the total OH reactivity) and OVOCs (on average 25 % of the total OH reactivity). The reactivity from isoprene makes a substantial contribution (often 20 %) to the total $k_{\mathrm{OH}}$ in the afternoon. Reactivity from alkanes and alkenes were dominated by small alkenes, mostly ethene and propene.

The median diurnal profile of the total OH reactivity had a maximum late at night. It decreased during the day by nearly 50 % and started to increase after sunset. Accumulation of OH reactants during the night could be due to fresh emissions that are released into the shallow nocturnal boundary layer. A similar diurnal profile was also observed for contributions from $NO_x$, alkane and alkene species. Their concentrations are typically connected to emissions from anthropogenic activities. OH reactivity from $NO_x$ was also the largest contribution to $k_{\mathrm{OH}}$ during night and early morning (20 to 30 %). The diurnal profile of $NO_x$ appears as the major driver for the diurnal profile of the entire $k_{\mathrm{OH}}$, whereas nearly all other contributions exhibited a less distinct diurnal profile. A different diurnal behavior to that for $NO_x$ was observed for isoprene, which is emitted by plants. The emission strength scales with light and temperature and, therefore, maximum mixing ratios were reached in the afternoon. Isoprene also contributed to the reactivity in the early evening most likely because isoprene that was emitted during daytime was only partly oxidized by OH before sunset. The diurnal profile of isoprene counteracted partly the decrease of OH reactivity due to the decrease of $NO_x$, alkane and alkene species.

CO mixing ratios ranged between 300 and 1000 ppbv during this campaign. Therefore, reactivity from CO made always a large fraction of the total $k_{\mathrm{OH}}$. The OH reactivity from CO showed only a weak diurnal variation with a median value of $3\,\mathrm{s}^{-1}$ and could therefore be used as indicator for the overall origin of pollutants apart from diurnal changes. As discussed above, measured $k_{\mathrm{OH}}$ scaled with CO indicating that also co-emitted OH reactants such as alkenes were important (Fig. 3).

A number of oxygenated volatile organic compounds (OVOCs) were measured in this campaign (Table 1). These included formaldehyde, acetaldehyde, glyoxal, methyl-vinyl ketone and methracrolein. Their reactivity made a large fraction of the total reactivity with median values between 2 and $4\,\mathrm{s}^{-1}$ over the course of one day. The largest contributions to the reactivity from OVOCs (more than $50\,\%$) came from formaldehyde and acetaldehyde (20 to $25\,\%$), while reactivity from other measured OVOCs such as acetone and glyoxal made only small contributions. These species can also originate from primary emissions. The good agreement between measured and calculated OH reactivity indicates nevertheless that these species were the most important organic oxidation products that contributed to the OH reactivity.

The reactivity of measured OVOCs shows a weak diurnal variation, with a decrease by a factor of about two from the morning to the evening. This behavior suggests that during daytime dilution due to a raising boundary layer height or chemical removal had a stronger influence on the observed OVOCs than fresh production by photochemistry.

Although the general behavior of OH reactivity and OH reactants was similar during the entire campaign, there were distinct differences in the magnitude of total OH reactivity during the first (7 June to 19 June) and second half (20 June to 8 July) of the campaign (Fig. 5). Measured OH reactivity was on average lower after 20 June specifically during the second half of the night and early morning, when median values were higher than $25\,\mathrm{s}^{-1}$ before 20 June and 16 to $20\,\mathrm{s}^{-1}$ later. Afternoon values were only slightly less after 20 June compared to the first part of the campaign. This is reflected in a decrease in median OH reactant concentrations during the second part of the campaign. It is most prominently seen in median alkene and alkane concentrations during nighttime (Fig. 5). In contrast, isoprene concentrations increased faster in the morning and high afternoon concentrations persisted in the evening during the second part of the campaign. Air temperatures were generally a few degrees higher than during the first two weeks, so that temperature driven biogenic emissions could have been larger after 20 June. The largest fraction of higher OH reactivity observed in the first part of the campaign remains unexplained by OH reactant measurements. However, even during times when measured reactivity was higher than calculations from OH reactants, the gap is within the combined $2\,\sigma$ uncertainties: The $k_{\mathrm{OH}}$ calculated from OH reactants has a $1\sigma$ uncertainty of $\pm10\,\%$ to $\pm15\,\%$ depending on the relative distributions of reactants and the measured $k_{\mathrm{OH}}$ has a $1\sigma$ uncertainty of maximum $\pm10\,\%$ plus $+0.7\,\mathrm{s}^{-1}$ (Table 1).

The good agreement between measured and calculated OH reactivity is also demonstrated by the high linear correlation coefficient ($R^2 = 0.77$ for the entire data set and both subsets of data) between both values (Fig. 7). For the second part of the campaign a linear regression analysis yields a slope of 0.96 with a negligible intercept of $-0.33\,\mathrm{s}^{-1}$. As already discussed, missing reactivity was higher during the first part of the campaign, so that a regression analysis yields a higher slope of 1.7 with an intercept of $-4.2\,\mathrm{s}^{-1}$. The larger intercept is due to a slightly non-linear relationship between measured and calculated reactivity for this subset of data.

Largest differences of 5 to $6\,s^{-1}$ (approximately 20 %) between measured and calculated OH reactivity occurred during nighttime and early morning during the first two weeks of the campaign, when also NO concentrations were highest. This could hint that unmeasured OH reactants were co-emitted with nitrogen oxides in combustion processes. Unknown compounds causing the missing reactivity are the main reason for the higher observed OH reactivity in the first two weeks. Therefore, there is no clear further hint about the nature of missing reactivity during this period. Emissions of organic compounds from biomass burning may have not been detected during the first part of the campaign. During nighttime also nearby sources for OH reactants as indicated by the short duration of high reactivity could have contributed to the missing reactivity. In addition, undetected products from the oxidation by the nitrate radical could have been part of missing reactivity in the night.

Exceptionally good agreement is seen at nearly all times after 20 June in the time series as well as in the median diurnal profile (Fig. 2 and 6). The median value of missing reactivity is only $0.3\,s^{-1}$. Such good agreement is not expected due to the large number of possible OH reactants in the atmosphere (Goldstein and Galbally, 2007). Specifically the number of OVOCs that were measured in this campaign is rather small (see above) and additional reactivity from other oxidation products could be expected to contribute to the total OH reactivity.

The good agreement between measured and calculated $k_{OH}$ indicates that other oxidation products than measured were not significantly contributing to the OH reactivity at the measurement site. Therefore, concentrations of oxygenated organic compounds that are produced by model calculations but that were not detected were constrained to zero in calculations presented in our accompanying paper by Tan et al. (2016), in order to ensure that modelled OH reactivity is consistent with measurements. One explanation could be that the photochemical age of air masses was short and therefore, oxidation products could not accumulate. This could be the case for fresh emissions close to the measurement site. In addition, the uncertainty of OH reactant measurements (up to 20 % for single compounds) would allow that unmeasured oxidation products significantly contribute to the total OH reactivity.

## 3.3 Comparison with previous field campaigns

In our previous field campaigns in China 2006 in the Pearl-River-Delta, PRD, (Hofzumahaus et al., 2009; Lou et al., 2010; Lu et al., 2012) and Yufa close to Beijing (Lu et al., 2013), OH reactivity was considerably higher, but exhibited a similar diurnal profile. Maximum values were 40 to $50\,s^{-1}$ in the night and early morning during the PRD and Yufa campaigns and reached minimum values around $20\,s^{-1}$ in the afternoon. Absolute contributions from CO and $NO_x$ were comparable with contributions in Wangdu 2014, with slightly higher CO concentrations in Yufa 2006. However, contributions from measured VOC were significantly higher in both previous campaigns compared to the Wangdu campaign in 2014 explaining partly the higher reactivity in these campaign.

In both previous campaigns, measurements of OVOCs were completely missing and the measured OH reactivity was found to be about two times larger than the total reactivity of measured CO, $NO_x$ and hydrocarbons (Lou et al., 2010). The missing reactivity could be quantitatively explained by OVOCs which were simulated by a model from the photo-oxidation of the measured VOCs. The major modelled OVOCs were formaldehyde, acetaldehyde, MVK, MACR and some minor isoprene oxidation products, which together could explain 70 % of the missing reactivity (i.e., about one third of the total reactivity). In the Wangdu campaign, the calculated total OH reactivity was largely in agreement with the measured $k_{OH}$. This time, formaldehyde, acetaldehyde, MVK, MACR and glyoxal were directly measured and also accounted for one third of the total reactivity. These species were also the most important OVOC species in other campaigns in anthropogenic dominated environments such as in Beijing (Shao et al., 2009), London (Whalley et al., 2016) and Tokyo (Yoshino et al., 2012). This confirms the high relevance of these specific carbonyl compounds as reactants for OH in the polluted boundary layer.

The OH reactivities measured at the Wangdu site in the North China Plain show diurnal profiles that are comparable to those reported for other polluted environments all over the world (see review by Yang et al. (2016)). The total reactivities lie within the range of values observed during summertime at other locations that were mainly influenced by anthropogenic emissions like Nashville (Kovacs et al., 2003), New York (Ren et al., 2003), Houston (Mao et al., 2010) in the US, Tokyo in Japan (Chatani et al., 2009), Beijing (Williams et al., 2016) in China, Seoul in South Korea (Kim et al., 2016) and London (Whalley et al., 2016) in Great Britain. Also the shapes of the diurnal profiles were similar with peak values between $15\,s^{-1}$ and $50\,s^{-1}$ in the early morning and minimum values in the afternoon. Significantly higher morning values of $130\,s^{-1}$ were observed in Mexico City 2003 (Shirley et al., 2006). Here, as well as in Wangdu and other urban sites, the diurnal shape of $k_{OH}$ was strongly determined by the variation of anthropogenically emitted $NO_x$ and co-emitted VOCs.

Care has to be taken, if missing reactivity is compared between different campaigns, because the number of measured OH reactants used to calculate the reactivity can significantly differ (Lou et al., 2010; Yang et al., 2016, and ref. therein). For the measurements in Beijing (Williams et al., 2016) approximately 25 % of the measured reactivity remained unexplained, although oxygenated organic species were partly measured. Approximately 30 % of the reactivity measured in Nashville could not be explained, even if modelled organic compounds were taken into account. For the other campaigns in anthropogenic influenced areas, measured OH reactivity could be explained by either measured OH reactants alone (New York, this campaign) or if in addition product species from model calculations were included (Yufa, PRD, Tokyo, London).

### 3.4 Experimental OH budget

OH reactivity measurements can be used not only to quantify the possible contribution of unmeasured OH reactants, but also allows quantification of the total OH production rate. Because OH is short-lived, it reaches a steady state within seconds. Thus, the total OH production rate ($P_{OH}$) equals the total destruction rate ($D_{OH}$). $D_{OH}$ can be calculated as the product of $k_{OH}$ and the OH concentration:

$$D_{OH} = k_{OH} \times [OH] \tag{2}$$

This rate can be compared with the sum of production rates ($P_{OH}$) from known OH sources. In this campaign, OH production from HONO and $O_3$ photolysis, ozonolysis of alkenes, and radical recycling reactions of $HO_2$ with NO and ozone can be calculated from measurements:

$$P_{OH} = P_{OH}(h\nu + O_3) + P_{OH}(h\nu + HONO) + P_{OH}(HO_2 + O_3) + P_{OH}(HO_2 + NO) + P_{OH}(O_3 + \text{alkene}) \tag{3}$$

Potentially unknown OH sources can then be determined as the difference between $D_{OH}$ and $P_{OH}$. This was successfully applied for data from our previous field campaigns in China (Hofzumahaus et al., 2009) revealing significant unaccounted OH sources and in chamber studies (Fuchs et al., 2013, 2014; Nehr et al., 2014).

The time series of calculated OH production and destruction rates are plotted in Fig. 2 and median diurnal profiles of quantities that are required for this calculation in Fig. 8. Unfortunately, the data coverage of simultaneous measurements before 20 June (mostly due to missing radical measurements) is not sufficient to allow for an independent analysis of the first part of the campaign as done for the analysis of OH reactants. However, results do not change significantly, whether the first part is included in the median diurnal profiles that are discussed below or not.

Figure 9 shows the median diurnal profile of the OH destruction and production rates and their difference including an estimate of the accuracy of the calculated difference. The diurnal profile of the OH production rate was mainly driven by solar radiation as expected from the photolytic nature of primary radical production, which also determines the overall abundance of $HO_2$. During the daytime, the known OH production was dominated by the recycling reaction of $HO_2$ with NO reaching a maximum of about $10 \, \text{ppbv/h}$ shortly before noon. The relative contribution of primary OH production by either $O_3$ or HONO photolysis to the total OH production was increasing during the day to reach median maximum values of $1.2 \, \text{ppb/h}$ and $1.5 \, \text{ppb/h}$, respectively. The ozone photolysis exhibited a strong diurnal profile because both, solar radiation and ozone concentration had maximum values at noon and early afternoon. An OH production rate from HONO photolysis of 1 to $1.5 \, \text{ppb/h}$ persisted into the afternoon due to relatively high HONO concentrations measured throughout the day. The budget of HONO will be discussed in a separate paper, but it is clear that HONO production from the reaction of OH with NO cannot explain the high HONO concentrations

in the afternoon. Ozonolysis of alkene species made only a minor contribution to the OH production at all times. Only $C_2$ to $C_6$ alkene species were measured, so that ozonolysis reactions of undetected alkene species (potentially monoterpenes) could have additionally contributed to the OH production. However, the good agreement between measured and calculated OH reactivity does not hint that a large fraction of alkene species were missed.

The time series of the total OH production and destruction rates, determined by Eq. 2 and 3, respectively, were nearly balanced for most of the time (Fig. 2). The OH destruction rate is on average only 20 % higher than the sum of OH production during daytime. Although the difference is hardly significant with respect to the experimental accuracies (Fig. 9), a systematic trend of the ratio between OH production and destruction rates with NO can be seen (Fig. 10), which points to a missing OH source at low NO concentrations.

For NO mixing ratios of less than 0.3 ppbv, OH destruction was nearly twice as large as the OH production, whereas production and destruction was balanced for NO mixing ratios higher than 1 ppbv. The result of the budget analysis is consistent with the finding by Tan et al. (2016) that model calculations underpredict OH by up to a factor of two at NO mixing ratios of less than 0.3 ppbv, but describe $HO_2$ and $k_{OH}$ correctly under these conditions at the Wangdu site. The good description of $HO_2$ and $k_{OH}$ means that the major known OH source (the reaction of $HO_2$ and NO) and the total OH loss rate are well represented by the model. Further model tests suggest a missing process that recycles OH from $RO_2$ and $HO_2$ by an unknown agent that behaves like 0.1 ppbv NO (Tan et al., 2016). Other trace gases measured at Wangdu give no hint to the nature of the missing source in the OH budget analysis or in the model results. A similar behavior was found in our previous field campaigns in China in 2006. However, the ratio of $P_{OH}/D_{OH}$ was much smaller, with a value of about 0.25 for NO mixing ratios of 0.1 to 0.2 ppbv NO in PRD (Hofzumahaus et al., 2009). In this case, the missing OH source was highly significant with respect to the experimental uncertainties of the calculated reaction rates, whereas in Wangdu, the much weaker imbalance of the OH budget can be almost explained by the experimental errors.

In addition to the measurement uncertainties stated in Tab. 1, instrumental tests during this campaign cannot exclude that OH concentration measurements are partly affected by an artifact as discussed in detail in Tan et al. (2016). The upper limit for an instrumental interference was estimated to be equivalent to an OH concentration of $1 \times 10^6$ cm$^{-3}$. This positive bias would also give a positive bias in the calculated OH destruction rate.

In the night, OH production from sources taken into account in this calculation is close to zero because there is no radiation. This suppresses both OH production from photolysis reactions and OH regeneration by the reaction of peroxy radicals with NO. Because of the relatively high OH reactivity OH concentrations are expected to be very small. However, median measured OH concentrations ranged between 0.5 to $1 \times 10^6$ cm$^{-3}$ (Fig. 8). A median OH production of 1 to 3 ppbv/h would be required to explain measured nighttime OH concentrations (Fig. 9).

Potential reasons for additional OH production at night have been recently discussed by Lu
et al. (2014), such as OH production by ozonolysis of terpenoids or dissociation of radical reservoir species like PAN that may be transported downward in the nocturnal boundary layer. Such mechanisms may have played a role at Wangdu, but we have no suitable measured data to test these hypotheses. In order to balance the calculated OH destruction rate during nighttime, a rather large concentration of an alkene would be required. Assuming an ozone concentration of 30 ppbv, a reaction rate constant for the ozonolysis reaction of $1.8 \times 10^{-15}\,\mathrm{cm^3 s^{-1}}$ for $\delta-$terpine and an OH yield of one (Atkinson and Arey, 2003), the concentration would need to be around 600 pptv.

However, the impact of a potential interference in the OH concentration measurements would also be largest in the night (Fig. 9), because nearly the entire OH signal could be due to interferences. As a consequence, the difference between calculated OH production and destruction during nighttime is within this additional uncertainty. The calculated OH destruction rate is less affected during daytime, when a potential OH interference of less than $1 \times 10^6\,\mathrm{cm^{-3}}$ would only be a small fraction of the total measured OH (Tan et al., 2016).

In our previous field campaigns in China 2006, the OH destruction and production rates were significantly higher than in this campaign. In PRD and Yufa, maximum mean turnover rates (OH destruction rates) of 40 ppbv/h and 20 ppbv/h, respectively, were reached around noontime (Lu et al., 2012, 2013). These values are 1.5 to 3 times higher than median OH turnover rates in this campaign. As discussed above, the major difference is that measured OH reactivities were significantly higher in the previous campaigns. The resulting higher loss rate was only partly balanced by a higher OH production from the reaction of $HO_2$ with NO, which was nearly a factor two larger in PRD and Yufa. Therefore, also the gap between calculated OH destruction and production was clearly above the level of significance with respect to the measurement uncertainties (Hofzumahaus et al., 2009).

Also the distribution of primary OH sources is different in this campaign compared to our previous campaigns in China, when HONO photolysis exhibited a diurnal profile with maximum values in the morning. These values were larger compared to this campaign, but HONO mixing ratios dropped to lower values in the afternoon, so that production by HONO photolysis was less in Yufa and PRD than in Wangdu 2016. Nevertheless, total primary OH production was higher (factor of 2 in PRD and factor of 1.5 in Yufa) in the previous campaigns.

HONO photolysis was also the most important primary source for OH radicals in other campaigns that were conducted in anthropogenic dominated environments for example in New York (Ren et al., 2003), in Paris (Michoud et al., 2012), Mexico City (Dusanter et al., 2009), Santiago (Elshorbany et al., 2009), and Tokyo (Kanaya et al., 2007). These campaigns took place in or very close to very large cities (the one in Paris during wintertime) and NO concentrations were often exceptionally high, so that HONO formation was favored. Our measurement site in Wangdu was not directly located in an urban area and therefore the $NO_x$ concentrations were only moderately high in the

morning and rather small in the afternoon, so that the importance of HONO as largest primary source for OH was not necessarily expected. The contribution of alkene ozonolysis to the OH production in other campaigns in urban environments were partly significantly higher (Kanaya et al., 2007; Dusanter et al., 2009; Elshorbany et al., 2009) compared to the Wangdu site due to higher alkene concentrations.

## 4 Summary and conclusions

OH reactivity was measured during a comprehensive field campaign at Wangdu in summer 2014. Additional measurements of OH reactants, OH concentrations and quantities that are required to calculate OH production ($HO_2$, NO, $O_3$, HONO, photolysis frequencies) allowed comparing OH reactivity measurements with calculations from measured OH reactants and analyzing the chemical OH budget from measurements alone.

Overall, measured OH reactivity can mostly be explained by OH reactants measurements, specifically during the second half of the campaign. Highest missing reactivity of the median diurnal profile (approximately 25 %) was observed during nighttime of the first part of the campaign, which could have been related to nearby emissions or undetected oxidation products. The diurnal profile of OH reactivity, the distribution of OH reactant and the good correlation of the OH reactivity with CO indicates that the chemical composition at the measurement site was mainly impacted by anthropogenic emissions. In our previous field campaigns in China 2006, the number of OH reactants that were measured was less and, thus, only approximately 50 % of the measured OH reactivity was explained by measured OH reactants (Lou et al., 2010; Lu et al., 2012, 2013). However, additional OH reactants determined by model calculations could close the gap in these cases. In this campaign, the good agreement between measured and calculated reactivity indicates that most important organic compounds were measured including oxidation products.

OH production and destruction were mainly balanced within the uncertainty of measurements. The accuracy of this calculation was lowered by additional uncertainty in the OH concentration measurements due to a potential bias (Tan et al., 2016). Despite this uncertainty, the OH destruction tends to be higher than OH production in the late afternoon, when NO concentrations were lowest. This result is consistent with the analysis of model calculations (Tan et al., 2016) and findings in previous field campaigns (Hofzumahaus et al., 2009).

However, in 2006 the observed discrepancy between the OH production and destruction rates was significantly larger requiring an additional OH source to close the gap. The major difference to this campaign was that the measured OH reactivity was much higher. Therefore, a significant gap in OH production and destruction rates were found in contrast to results in this campaign. For future field work, comprehensive studies like this campaign in photochemically active environments where larger contributions from biogenic reactants can be expected in addition to anthropogenic emissions

may help to solve the still open questions of imbalances in the OH production and destruction and measured and calculated OH reactivity that have been observed in other campaigns.

*Acknowledgements.* We thank the science teams of Wangdu-2014 campaign. This work was supported by the National Natural Science Foundation of China (Major Program: 21190052 and Innovative Research Group: 41121004), the Strategic Priority Research Program of the Chinese Academy of Sciences (grant no. XDB05010500), the Collaborative Innovation Center for Regional Environmental Quality, and by the EU-project AMIS (Fate and Impact of Atmospheric Pollutants, PIRSES-GA-2011-295132). The authors gratefully acknowledge the NOAA Air Resources Laboratory (ARL) for the provision of the HYSPLIT transport and dispersion model and/or READY website (http://www.ready.noaa.gov) used in this publication.

595

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

**Table 1.** Instruments deployed in the campaign and used for data analysis.

| | measurement technique | time resolution | $1\,\sigma$ detection limit | $1\,\sigma$ accuracy |
|---|---|---|---|---|
| $k_{\mathrm{OH}}$ | LP-LIF [a] | 180 s | $0.3\,\mathrm{s^{-1}}$ | $\pm 10\,\% +0.7\,\mathrm{s^{-1}}$ |
| OH | LIF [b] | 32 s | $0.32 \times 10^{6}\,\mathrm{cm^{-3}}$ | $\pm 11\,\%$ |
| HO$_2$ | LIF [b] | 32 s | $0.10 \times 10^{8}\,\mathrm{cm^{-3}}$ | $\pm 16\,\%$ |
| photolysis frequency | spectroradiometer | 20 s | [c] | $\pm 10\,\%$ |
| O$_3$ | UV photometry | 60 s | 0.5 ppbv | $\pm 5\,\%$ |
| NO | chemiluminescence | 180 s | 60 pptv | $\pm 20\,\%$ |
| NO$_2$ | chemiluminescence[d] | 600 s | 300 pptv | $\pm 20\,\%$ |
| HONO | LOPAP[e] | 300 s | 7 pptv | $\pm 20\,\%$ |
| CO, CH$_4$, CO$_2$, H$_2$O | cavity ring down | 60 s | [f] | [g] |
| SO$_2$ | pulsed UV fluorescence | 60 s | 0.1 ppbv | $\pm 5\,\%$ |
| HCHO | Hantzsch fluorimetry | 60 s | 25 pptv | $\pm 5\,\%$ |
| volatile organic compounds[h] | GC-FID/MS [l] | 1 h | 20 to 300 pptv | $\pm 15$ to $20\,\%$ |
| volatile organic compounds[i] | PTR-MS | 20 s | 0.2 ppbv | $\pm 15\,\%$ |
| glyoxal | CEAS[j] | 1 s | 0.02 ppbv | $\pm 5.8\,\%$ |

[a] laser photolysis - laser induced fluorescence

[b] laser induced fluorescence

[c] process specific, 5 order of magnitudes lower than maximum in noon time

[d] photolytical conversion to NO before detection, home built converter

[e] long-path absorption photometry

[f] species specific, for CO: 1 ppbv; CH$_4$:1 ppbv; CO$_2$: 25 ppbv; H$_2$O: 0.1 % (absolute water vapor content);

[g] species specific, for CO: 1 ppbv; CH$_4$:$\pm$1 ppbv; CO$_2$: $\pm$25 ppbv; H$_2$O: $\pm$5 %

[h] VOCs including C$_2$-C$_{11}$ alkanes, C$_2$-C$_6$ alkenes, C$_6$-C$_{10}$ aromatics

[i] OVOCs including acetaldehyde, methyl-vinyl ketone and methacrolein

[j] cavity-enhanced-absorption spectroscopy

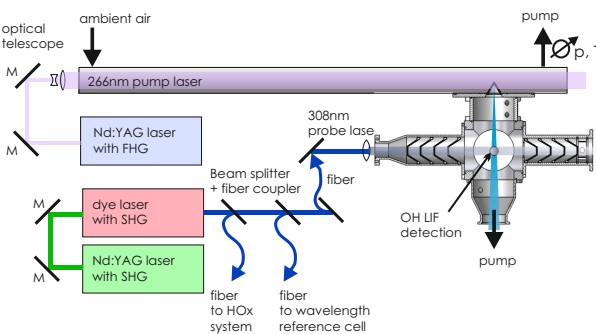

**Figure 1.** Schematics of the Jülich OH reactivity instrument (M: turning mirror). Ambient air is sampled into a flow tube. A small part of the air is drawn into the OH detection cell that is operated at a pressure of $4\,\mathrm{hPa}$. High OH concentrations are produced by flash photolysis of ozone at $266\,\mathrm{nm}$ at a low frequency of 1 to $2\,\mathrm{Hz}$. The OH concentration is probed at a high frequency of $8.5\,\mathrm{kHz}$, so that the loss of OH radicals due to their reaction with OH reactants in the ambient air can be observed.

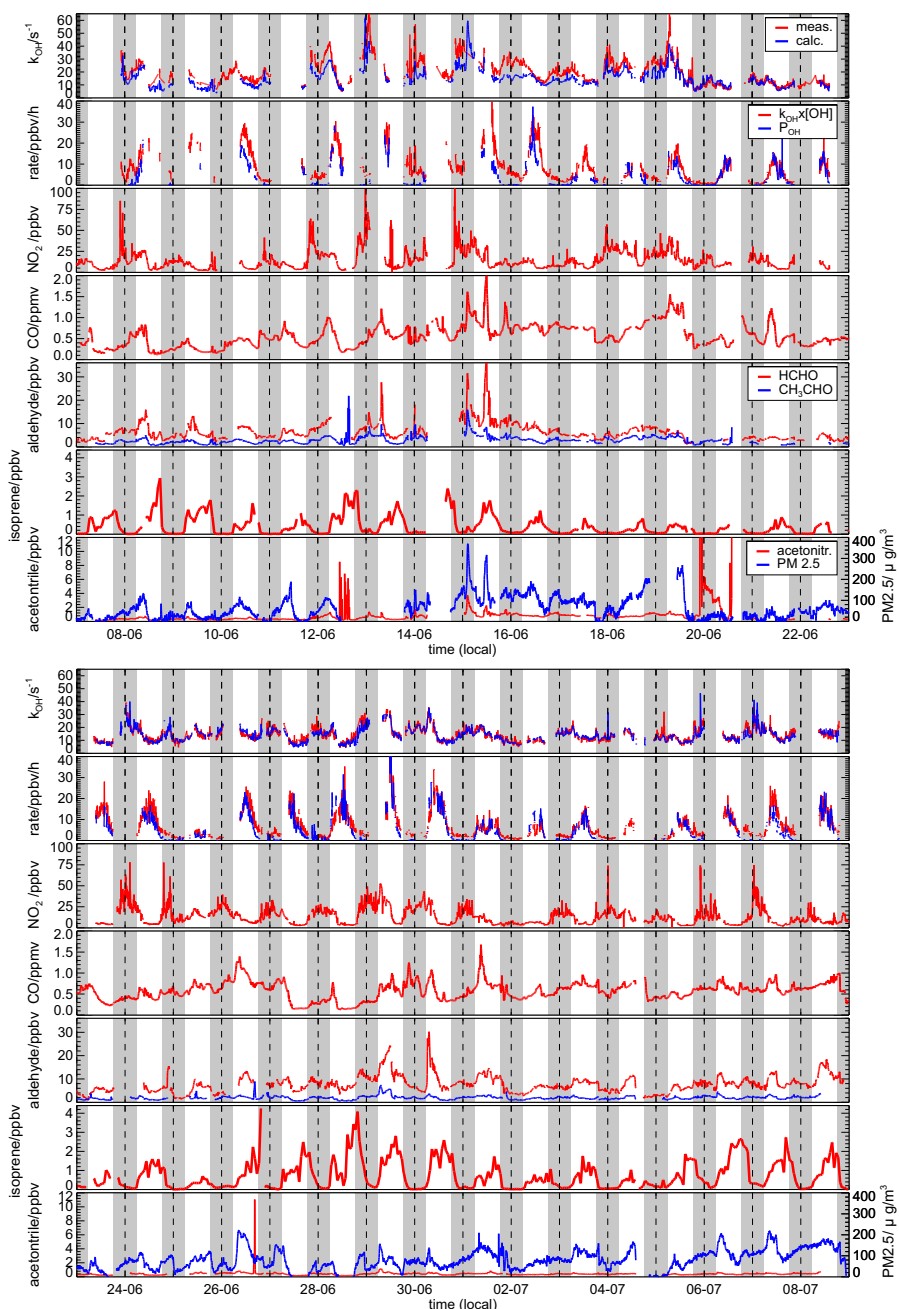

**Figure 2.** Time series of measured and calculated OH reactivity. In addition, time series of the OH destruction rate ($D_{OH}$) calculated from measured OH concentrations and OH reactivity is shown together with the sum of measured OH production rates ($P_{OH}$) from $O_3$ and HONO photolysis and reactions of $HO_2$ with NO and $O_3$. Lower panels give time series of important trace gas measurements contributing to the OH reactivity. Gray areas indicate nighttime.

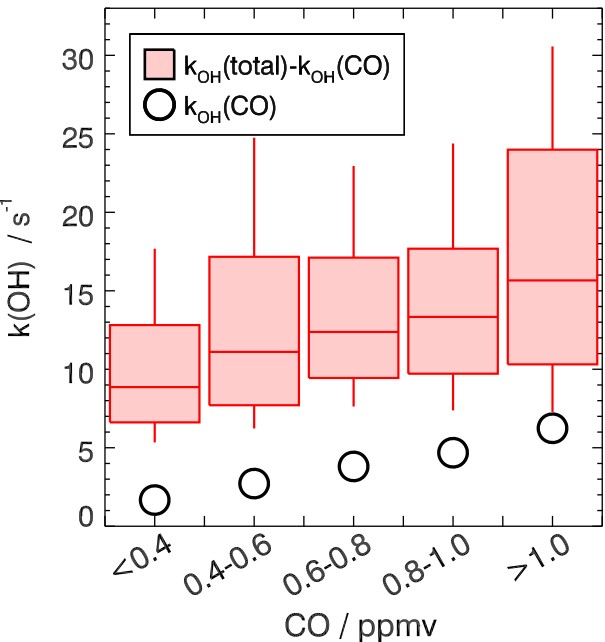

**Figure 3.** Correlation between OH reactivity excluding CO and CO mixing ratios. Red boxes give 25 and 75 percentiles and whiskers 10 and 90 percentiles of the $k_{OH}$ distribution. Black circles show median values of OH reactivity that is caused by CO.

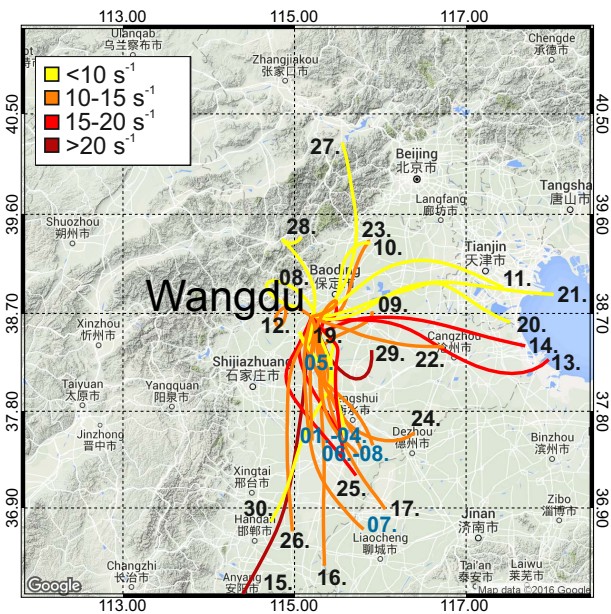

**Figure 4.** NOAA Hysplit 24-hour back-trajectories during the campaign calculated as averages of hourly back-trajectories between 10:00 and 19:00 local time. Colors of trajectories indicate the OH reactivity level measured at the field site in Wangdu. Black numbers indicate the date in June, dark blue numbers the date in July.

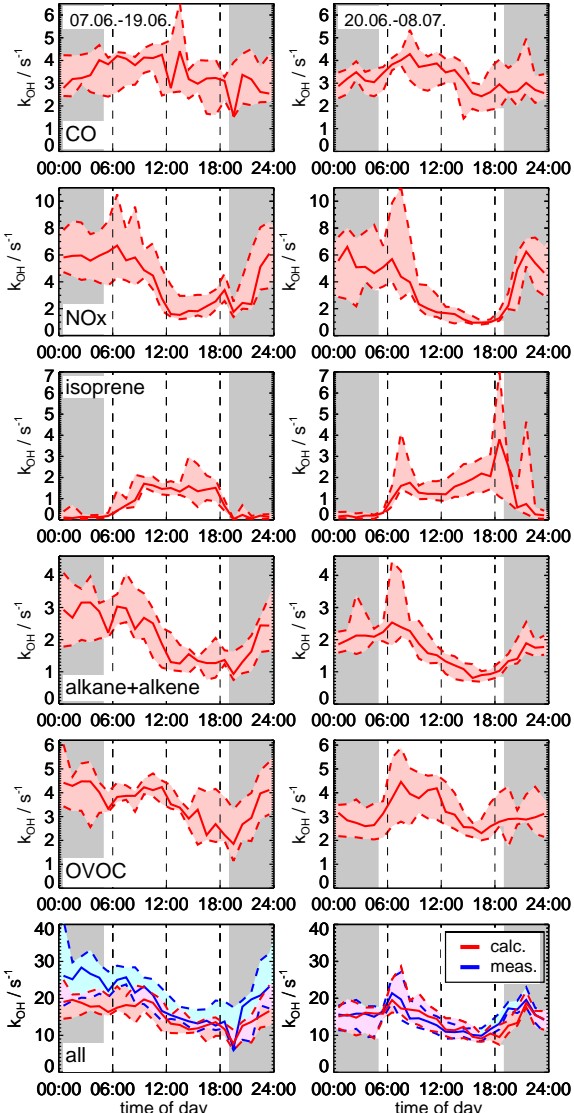

**Figure 5.** Median diurnal profiles of reactivity from major measured OH reactants and of the total measured and calculated OH reactivity for the first and second part of the campaign. Data is only included, if all major OH reactants and OH reactivity were concurrently measured. Colored areas give 25 and 75 percentiles. Gray areas indicate nighttime.

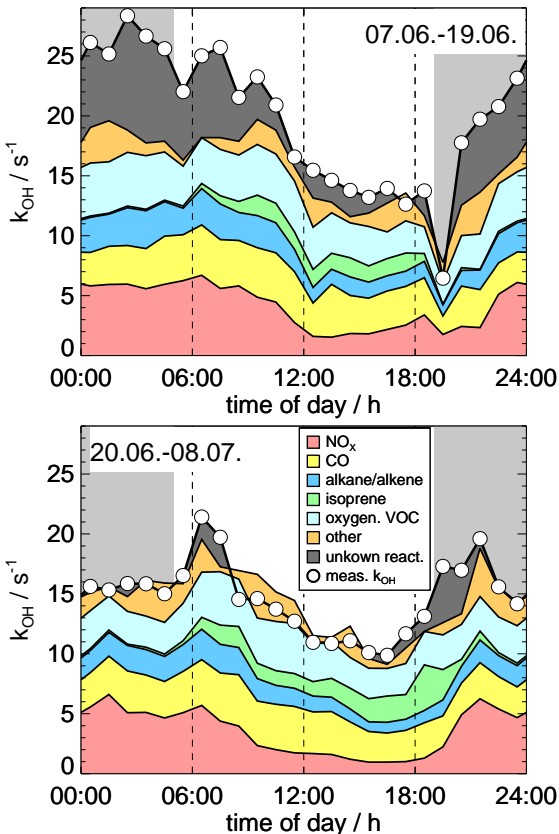

**Figure 6.** Sum of median diurnal profiles of reactivities from all measured OH reactants compared to the measured OH reactivity for the first and second part of the campaign. Data is only included, if all major OH reactants and OH reactivity were concurrently measured. "Other" include small contributions from measured OH reactants listed in Table 1 ($CH_4$, $SO_2$, aromatics). The dark grey area indicates missing OH reactivity from unmeasured OH reactants. Light gray areas indicate nighttime.

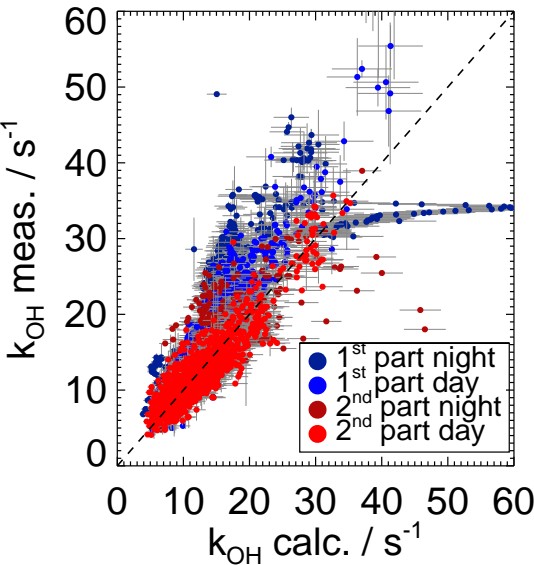

**Figure 7.** Correlation between calculated and measured OH reactivity with color coded periods. The linear correlation coefficient ($R^2$) is 0.77 for the entire data set and both data from both periods alone.

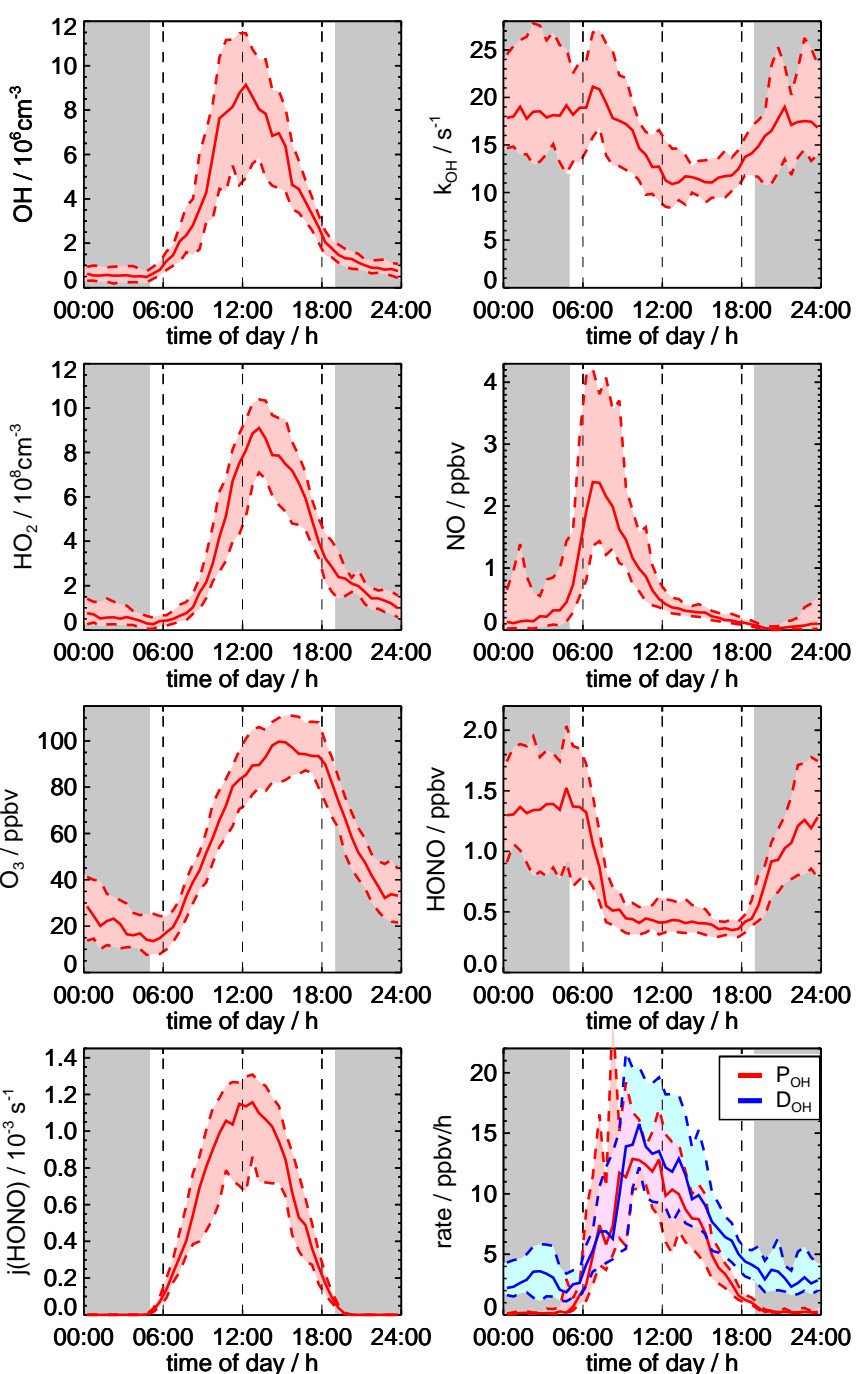

**Figure 8.** Median diurnal profiles of trace gas concentrations used for the calculation of the total OH production rate ($P_{OH}$) and destruction rate ($D_{OH}$). Data is only included, if all required trace gas concentrations and OH reactivity were concurrently measured. Colored areas give 25 and 75 percentiles. Gray areas indicate nighttime. Note that the selection of data is different for median profiles shown in our accompanying paper by Tan et al. (2016).

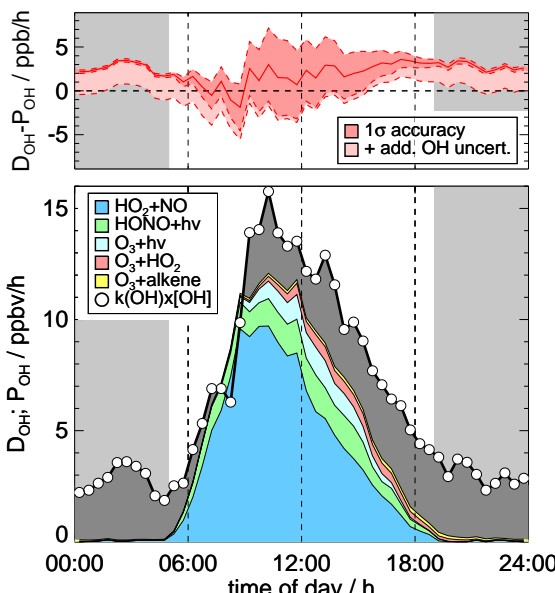

**Figure 9.** Median diurnal profiles of OH production ($P_{OH}$) and destruction ($D_{OH}$) rates. Data is only included, if all required trace gas concentrations and OH reactivity were concurrently measured. Dark grey areas indicate missing OH production. The upper panel gives the $1\,\sigma$ accuracy of the difference ($D_{OH}$- $P_{OH}$) calculated from the uncertainties of measurements (Gaussian error propagation). The effect on the accuracy from an upper limit of potential interferences in the OH measurements is shown separately.

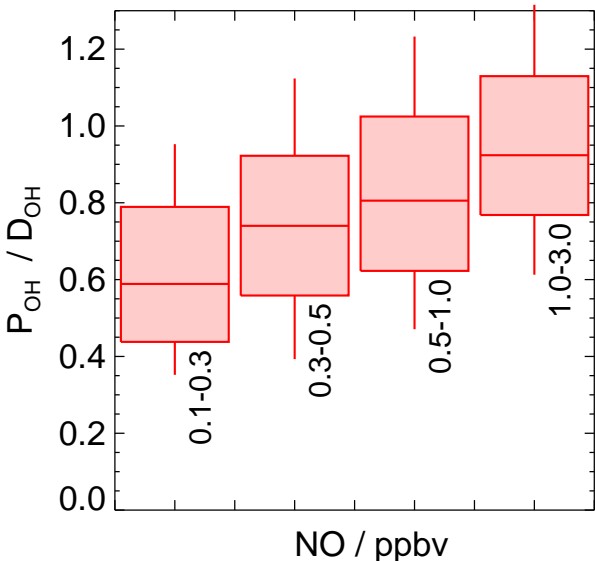

**Figure 10.** Box and whisker plot of the ratio of the total OH production ($P_{OH}$) and the OH destruction rate ($D_{OH}$) as a function of the NO mixing ratio for daytime values. Boxes give 25 and 75 percentiles and whiskers 10 and 90 percentiles. Data is only included, if all required trace gas concentrations and OH reactivity were concurrently measured. Gray areas indicate nighttime.