# Peer review of "OH reactivity at a rural site (Wangdu) in the North China Plain: Contributions from OH reactants and experimental OH budget"

_Atmospheric Chemistry and Physics, 2016_

## Referee Comment (RC1) · Anonymous Referee #1 · 14 Sep 2016

Fuchs et al. present direct measurements of OH reactivity performed during a comprehensive field campaign conducted in June-July 2014 close to the small town of Wangdu. After a description of OH reactivity measurements and a brief description of radicals and trace gases measurements, the authors discuss the OH reactivity time series and compare the OH reactivity measurements with calculated OH reactivity estimated using measured OH reactant concentrations. This comparison shows, generally, good agreement between calculated and measured OH reactivity. Finally, the authors analyze the OH budget and perform a comparison between total OH production and destruction rates to quantify unaccounted OH production. This highlights an imbalance in the late afternoon and at night when NO concentrations were lowest.

This manuscript is within the scope of ACP and will be of interest for the atmospheric community. I therefore recommend publication in ACP after the authors address the comments listed below.

Main comments:

1) Although a complete description of OH reactivity measurements is performed, the descriptions of ancillary measurements are not sufficiently detailed. Even if these measurements are described elsewhere (Tan et al., ACPD, 2016), the section 2.2 is too short and description of different measurements, especially for NO, HONO and VOCs (both GC-FID/MS and PTR-MS measurements), should be given in more details. For example: What are the model and brand of instruments? What were the frequencies of calibration for the various measurements? How were they performed? How many VOCs were measured by different instruments?

Authors report only acetaldehyde and sum of MVK and MACR as species measured by PTR-MS only, other species measured by PTR-MS being also measured by GC system (isoprene, benzene, toluene, styrene, C8-aromatics, C9-aromatics) (see P5, line 98-103). If so, PTR-MS measurements seem under-exploited (see de Gouw and Warneke, Mass Spectrom. Rev., 26, 223–257, 2007, for a review). Did you really measure so few compounds with PTR-MS during the campaign? If more compounds were measured by PTR-MS, it should be clarified in the section 2 of the manuscript.

Furthermore, no description of NO2 or photolysis frequency measurements is made in the section 2, while these measurements are used for estimation of calculated OH reactivity and OH production rate, respectively.

P4, line 85: "Nitrogen monoxide was also detected by several instruments". Please indicate how many instruments measured NO as well as their model and brand.

P4, line 86: "Measurements from one of the instruments". Please detail which instrument it is.

P5, line 89-90: "Nitrous acid (HONO) concentrations were simultaneously measured by several instruments applying different measurement techniques". Please specify which instruments were used to measure HONO (brand, model, technique).

P5, line 90-91: "The agreement between instruments was diverse". Please develop this statement.

2) While the authors observed an imbalance between total OH production and OH destruction rates, especially in the late afternoon and at night when NO concentrations are low, only few hypotheses, from literature, are given to explain it. It would be interesting to investigate further this observed discrepancy to identify potential unaccounted OH sources in POH calculations in Wangdu.

P13, line 399-400: "Ozonolysis of alkenes species made only a minor contribution to the OH production at all time". This is not necessarily expected in anthropogenic dominated environments where these reactions can represent an important fraction of OH production in the late afternoon and at night (e.g. Ren et al., Atmos. Environ., 37, 3639-3651, 2003; Kanaya et al., J. Geophys. Res., 112, 2007; Dusanter et al., ACP, 9, 6655–6675, 2009), precisely the time period when the largest imbalance between POH and DOH is observed. How many and which alkenes were measured? Is it possible that an underestimation of the contribution of ozonolysis of alkenes in OH production rate, due to unmeasured alkenes, is, at least partly, responsible for the discrepancy observed between POH and DOH in the late afternoon and at night?

Figure 2: Large discrepancies are observed between DOH and POH on 10 and 15 June. Maybe these days could be studied in more details to investigate potential missing OH source. At least, the large imbalance between OH production and destruction rates observed these two days could be discussed in the manuscript.

Minor comments:

P3, line 51-52: "from ozone and nitrous acid photolysis and reaction of hydroperoxyl

radicals with nitric oxide": Please add reactions of ozonolysis of alkenes and reaction of hydroperoxyl radical with ozone to these OH sources list to be exhaustive.

P4, line 85: "nitrogen monoxide" is used to define NO while the term "nitric oxide" is used elsewhere in the manuscript. Please harmonize.

P6, line 136: "nitrogen oxides" to "nitric oxide". Ozone depletion at night is due to its titration by NO.

P7, line 181: Typo: "(??)"

P7, line 181-182: "could not fully excluded" to "could not be fully excluded".

P7, line 193: "In general, values were lower during daytime than at night". Please indicate daytime and nighttime median values of OH reactivity.

P9, line 240: "is not necessarily expected" to "is not expected".

P9, line 246: Please indicate how many and which species are considered in the estimation of calculated OH reactivity? What are the reaction rate constants used? All these information could be given, for example, in a table in supplementary material. These information are important to estimate the representativeness of missing OH reactivity.

P9, line 247-249: "Because of the similarity of diurnal profiles of observations during the first and the second part of the campaign, measured kOH and calculated reactivity from major contributors are shown as median diurnal profiles with percentiles in Fig. 5". I do not understand this statement since median diurnal profiles of the first and the second part of the campaign are presented separately in figure 5. Please clarify.

P10, line 279: "Only relatively few oxygenated volatile organic compounds". Please indicate how many OVOCs were measured.

P10, line 284: "the most important organic oxidation products". This statement should be moderated or removed since formaldehyde and acetaldehyde are also emitted directly in the atmosphere by anthropogenic and biogenic sources (e.g. Chen et al., ACP, 14, 3047-3062, 2014).

P11, line 310-313: "Largest differences of 5 to 6 s−1 (approximately 20 %) occurred during nighttime and early morning during the first two weeks of the campaign, when also nitrogen oxide concentrations were highest. This could hint that unmeasured OH reactants were emitted concurrently with nitrogen oxides in combustion processes". Can you correlate the missing reactivity to several source tracers (e.g. NOx, Acetonitrile etc. . .) trying to identify the nature of the OH reactants responsible for missing reactivity, especially during the first part of the campaign?

P11, line 315: "Exceptionally good agreement is seen at nearly all times after 20 June". Please be more quantitative.

P11, line 317-318: "the number of OVOCs that were measured in this campaign is rather small". Please indicate how many OVOCs were measured.

P11, line 322: "the photochemical age of air masses was short". Can you make an estimate of photochemical age of air masses during the campaign to support this statement?

P13, line 395: typo: "vales" to "values".

P14, line 422-423: "NO that is mainly formed from NO2 photolysis". The main source of NO is its emission from combustion processes. Please rephrase or remove this statement.

P15, line 464: typo: "NO O3" to "NO, O3"

Figure 2: Please replace "ppm" by "ppmv" for CO to be consistent with the other units.

Figure 2: High concentrations of isoprene (up to 4 ppbv for example on 26 and 28 June) are sometimes observed after sunset. What are the sources of isoprene at night? Could it be due to interferences? These high concentrations lead, in particular,

to large OH reactivity from isoprene in the late afternoon (after 18:00) and at night especially during the second period of the campaign (see Figure 5).

Figure 6: Please define which species exactly are considered in the "other" group.

Figure 8: Dark grey area should also be defined in the legend.

[Figure]

---

## Referee Comment (RC2) · Anonymous Referee #2 · 15 Sep 2016

This paper describes and analyses the OH reactivity measurements carried out at the Wangdu site, denominated as "rural site" in the title. The aim of this work is to improve the knowledge of radical chemistry in Chinese megacity area and complete the analysis of this chemistry at the Wangdu site based on the comparison between measured and modeled concentration of OH, HO2 and RO2 radical, described in an accompanying paper (Tan, 2016). In the present paper, the site and the instrumentation used for ancillary measurements are briefly described. The newly built OH reactivity instrument (similar to the one used in previous campaigns in China) and the fitting procedures are described. The results are presented as follows:

- first a general description of the time series of OH reactivity, including 2 different

[Figure]

periods : one with higher reactivities, related to local biomass burning activities, and one at lower reactivities ,

-then a comparison between measured and calculated reactivity on the basis of ancillary measurements including for the first time some OVOCs, contributing to a large fraction of the OH reactivity. In the second period of the campaign (low reactivity), the agreement is good, but during the first part of the campaign, for some periods, during the night and at high nitrogen oxide concentrations, unexplained missing reactivity is found

-a discussion based on the OH reactivity measurement in other campaigns in China and at some locations influenced by anthropogenic emission is done,

- finally, an analysis of the OH budget based on the calculation of the production and destruction rate is done using OH reactivity, HOx, ozone, HONO, NO, photolysis frequencies and alkenes measurements. This analysis highlights a production rate slightly larger than the destruction rate the afternoon, this difference decreases with increasing NO concentration, as observed, more clearly in previous field campaigns in China. Comparisons of the conditions point out the differences with the present campaign with lower NO (factor 2), lower reactivity (factor 1.5-3), different HONO profiles.

The article is well written and is suitable for publication in ACP however, I would suggest to add :

- more connections with the conclusions from the accompanying paper (Tan, 2016) in the present paper, particularly because the modeled OH reactivity is presented in (Tan, 2016) as well as sensitivity run constrained on the basis of OH reactivity measurements,

- a more detailed discussion on the missing reactivity,

- gathering the discussion on the comparison with previous campaigns to avoid repetition and to help the reader to better see the new understanding brought by this

campaign and in particular the reactivity measurements.

L29 : It is mentioned in the introduction that " the effort to improve our knowledge of radical chemistry in Chinese megacity areas was continued by a comprehensive field campaign at a location close to the city Wangdu" please resume the conclusion of the accompanying paper (Tan, 2016) and put in the conclusion of the article the key results from this campaign which contribute to this improvement. What type of environment should be studied in future campaigns to bring complementary information as in the present study the low reactivity and potential OH interferences seem to prevent to draw clear conclusion concerning the OH budget ?

It is confusing to see in the introduction that the site is described as "close to the city Wangdu", described at rural in the title and that the campaign location is in a botanic garden close to the "small" town Wangdu. Please clarify how and why this site has been chosen and how it is classified.

Linked comment : how the comparison with other campaigns has been chosen ? L350-365, related to anthropogenic emissions but without the comparisons with reactivity measurements done in Paris or Mexico, whereas these campaigns are discussed later (l453-460) concerning HONO. I would propose to gather the comparison paragraphs.

L73 : Particle measurements are also available. Could it be commented ?

L156 : how the delay to start the fit is defined (which level of deviation is considered to discard the points ?)

L157 : the reason for the deviation is attributed to the non homogeneity of the OH distribution: could this be clarified? Is it due to the heterogeneous distribution of the laser energy ? L165 : which species considered as contaminations have been identified in the gas cylinder ?

L184 : why the assumption of a bi-exponential fit would better describe the conditions with recycling ?

L189 : which criterion is used to decide that the measurements appear as single exponential decays ?

L206 : would be useful to show other correlations with individual species (provide also more details in the repartition of the reactivity for specific periods (L310)

L214/Figure 4 : difficult to identify the different trajectories

L232 : other species correlate with acetonitrile (could provide the profile in Figure 2) ? New peaks (GC, PTR-MS) have been observed during the biomass activities?

L264 and 302 : influence of products due to oxidation by nitrate radical could be discussed, parallel with particle profiles could be discussed.

L280-291 : some repetitions with 3.1

L297 : "It is most prominently seen in median alkene and alkane concentrations". Where ?

L322 : possible to use ratio of species with different rate constants with OH to estimate this photochemical age ?

L350-365 : see comment above. What can be concluded from these comparisons ?

L382 : What is missing ? How it has been concluded that "results do not change significantly, whether the first part is included or not"? Please detail.

L419 : possible to show the results with the bias subtracted ?

L432 : Please detail

L439 : why introducing the term turnover rates there ? Useful ? Check the consistency of the terms used in the Figure 8

L475 : but there are periods with significant missing reactivity. It could be mentioned in the conclusion.

---

## Referee Comment (RC3) · Anonymous Referee #3 · 19 Sep 2016

This manuscript describes measurements of OH reactivity made in Wangdu, China, using the laser photolysis with laser induced fluorescence (LP-LIF) technique, and makes comparisons with the calculated OH reactivity based on measurements of OH reactants. In general, the observed OH reactivity can be explained by measurements of OH reactants (which included OVOCs), with a limited role for unmeasured oxidation intermediates.

Measurements of OH radical concentrations, presented in a separate paper, are also used with the OH reactivity to assess understanding of the OH production rate at the site. The OH production and destruction were generally balanced throughout the campaign, although there is potential for a bias in the OH concentration measurements

which may impact the analysis of the OH production rate reported in the manuscript.

The paper is within the scope of ACP, and I recommend publication once a number of issues have been addressed. The analysis of the OH reactivity is generally well presented, but it is poorly quantified. There are numerous instances throughout the manuscript where claims are made regarding 'good correlations' or 'good agreement' between measurements and calculations but these statements must be quantified. The comparisons between measured and calculated OH reactivity are presented only as time series, the manuscript would benefit from a scatterplot showing the calculated OH reactivity against the measurements, enabling a more direct comparison.

The manuscript reports a limited role for oxidation intermediates. Model results reported for OH concentrations in the manuscript by Tan et al. could be used to quantify the contributions of unmeasured oxidation intermediates to the total OH reactivity, but are only briefly discussed.

In places, the manuscript is also poorly written and could be clarified significantly. Please see specific comments below.

Specific comments: Abstract: Mention the technique used to make the OH reactivity measurements. Lines 7-8: Quantify 'good correlation' and 'high contribution'. Line 15 (and elsewhere): 'hydroperoxy' is preferred over 'hydroperoxyl'. Line 20: 'oxidization' to 'oxidising'. Line 33: 'aerosols' to 'aerosol' or 'properties of aerosols'. Line 49: State the isoprene oxidation products that were measured. Line 63-69: Can you comment on the prevailing wind direction? Line 70: 'sea containers' to 'shipping containers' and perhaps replace 'partly stacked up' by 'raised'. Line 84: 'well agreed' to 'agreed well', and quantify the agreement (and elsewhere, e.g. line 102). Line 85: How many instruments? What was the standard deviation of the average? How did one of the instruments 'appear to be more precise'? How does the uncertainty in the NO measurements impact the analysis of OH reactivity? Line 93: How small is 'rather small'? Quantify the impact. Line 98: 'glyoxal'. Line 100: 'mass spectrometry'. Line 101: 'part

Interactive
comment

of the same species' – please rephrase to clarify the meaning. Line 113: Please clarify that it is the previously reported Zeppelin instrument that is being used in this campaign. Line 121: Does the length of the sampling line affect the measurements? Line 126: How does the change in temperature affect the measured OH reactivity? What is the difference from the external ambient temperature? Line 141: 'In a distance' to 'At a distance'. Line 150: Clarify what you mean by 'sufficiently precise'. How does the summing of decay curves affect the reported values? Does it make any difference to simply average 60 decays before fitting as opposed to summing ten decays and then averaging six summed decays? Line 153: 'equalizes' to 'equalize'. Line 165: What is the uncertainty/precision of the measurements? Line 181: '??'. Line 187: What was the mean/standard deviation/median NO? 'Thus' is mis-used, it doesn't necessarily follow that no bi-exponential behavior was observed because NO concentrations were low (i.e. high NO concentrations are not the only possible explanation for bi-exponential behavior). Line 203: Quantify the correlation. Lines 215-234: Much of this is poorly phrased and difficult to follow (particularly lines 224-228), please consider re-writing. Line 220: 'measurements' to 'measurement'. Line 229: 'are' to 'is'. Clarify what 'reason' is explaining. Line 239: Comma after 'OH reactants'. Line 243: Remove 'concentrations', or replace 'VOCs' with 'VOC' and remove 'species'. Line 260: Comma after 'small alkenes'. Line 262: 'in the late night' to 'late at night'. Line 269: Avoid use of 'opposite', it implies a more perfect mirror relationship than I expect was observed. Replace 'than' with 'to'. Line 274: 'range' to 'ranged'. Switching from one tense to another throughout the following section. Line 300: 'degree' to 'degrees'. Line 305: Remove the comma after 'times'. Line 308: Re-iterate the source of the uncertainties and why they have been separated into two terms. Line 310: What were the differences between? The wording of the sentence implies it could be between nighttime and early morning as opposed to measured and calculated reactivity. Line 315: Quantify the 'exceptionally good agreement'. Line 345: 'accounted also' to 'also accounted'. Line 355: Avoid 'like in this campaign', perhaps 'Similarly to observations in this campaign'. Line 364: The measured OH reactivity in London contained significant contributions from

model-generated intermediates. Line 384: Avoid use of 'like for'. Line 389: 'during the daytime'. Line 402 and following paragraph: Quantify 'nearly balanced', 'slightly larger' and 'hardly significant'. What was the 'systematic trend'? Line 411: Quantify 'much larger and highly significant'. Line 422: What about the possibility of OH regeneration through peroxy radical reactions with the nitrate radical? Line 431: 'hypothesis' to 'hypotheses'. Line 438: Define the term 'turnover rate'. Line 445: Quantify or define the level of significance in the term 'clearly above the level of significance'. Line 464: Comma after NO. Line 467: 'OH reactants' to 'OH reactant'. Line 475: 'all' to 'most'.

---

## Referee Comment (RC4) · Anonymous Referee #4 · 28 Sep 2016

Fuchs and colleagues presented a field campaign dataset in Wangdu China and mostly focused on interpreting ambient total OH reactivity dataset using a comprehensive trace gas dataset. They reported some occasional missing OH reactivity but much lower level than previously reported extreme cases. This may be a consistent finding in the anthropogenic dominant environments. Then, the comparisons with previous fieldworks are presented. Finally, the comparison between OH production rates and loss rates are compared and concluded that observationally assessed OH destruction rates are in general higher than the OH production rates. As the authors claimed that interests in megacity air quality especially countries like China have grown but field observational datasets have been very limited. Therefore, the presented dataset and

analysis will be highly beneficial to the research community. I am in favor of publication of this manuscript with the clarifications of a couple of points raised in the manuscript.

1. 305 – 310: Most of uncertainty analysis that I have encountered uses at least 2 sigma uncertainty rather than 1 sigma. If you use 2 sigmas probably the calculated OH reactivity would be able to account measured OH reactivity. 2. Figure 7 and Figure 8: It has been highly controversial about the nighttime OH. As observed, the lifetime of OH is much shorter during the nighttime so it is rather surprising to see observed night-time OH such a low OH production during the night. There is an obvious attempt to account the observation such as ozonolysis of terpenoids and dissociation of potential contributions from PANs but could not provide a quantitative assessment since there is no observation data. However, as it is so important issue, I think at least the authors should attempt to assess what kind of PANs or terpenoids levels you would need to account the night time OH. Otherwise, as the discrepancy between OH production rates and OH destruction rates are appeared almost identical except in the early morning, some may conclude that the discrepancy may be simply accounted by an instrument artifacts as described in the manuscript. 3. (minor comment) Recently, Kim et al (2016) reported observed OH reactivity in the Seoul Metropolitan Area. The addition of this reference to the comparison could be useful.

Reference: Kim et al., OH Reactivity in Urban and Suburban regions in Seoul, South Korea-An East Asia megacity in a rapid transition. Faraday Discussions DOI:10.1039/C5FD00230C, DOI:10.1039/C1035FD00230C (2016).

---

## Author Comment (AC1) · 24 Nov 2016

We thank the reviewer for the helpful comments.

**Comment:** Although a complete description of OH reactivity measurements is performed, the descriptions of ancillary measurements are not sufficiently detailed. Even if these measurements are described elsewhere (Tan et al., ACPD, 2016), the section 2.2 is too short and description of different measurements, especially for NO, HONO and VOCs (both GC-FID/MS and PTR-MS measurements), should be given in more details. For example: What are the model and brand of instruments? What were the frequencies of calibration for the various measurements? How were they performed? How many VOCs were measured by different instruments?

**Response:** We add more information about instrument models and brands.

We also add information about calibrations for (1) HONO measurements on p5 l93: "This instrument was calibrated by using a liquid standard as described in Li et al. (2014) every ten days." (2) NOx on p4 l86: "Daily calibrations were performed using a certified gas standard." (3) VOC measurements by GC on p5 l96: "Full calibrations using certified gas standards (Air Environmental, Spectra Gases) were done before and after the campaign. Drifts of the sensitivity during the campaign were accounted for by measuring the instrument sensitivity for bromochloromethane, 1,4-difluorobenzene, chlorobenzene, and 1-bromo-3-fluorobenzene every second day." (4) VOC detection by PTR-MS on p5 l105: "Calibration of the PTR instrument was done every day using a certified gas standard (Air Environmental Inc.)." (5) and HOx measurements on p4 l81: "The instrument sensitivity was calibrated every 3 to 4 days by a custom-built calibration source described in detail in Fuchs et al. (2011)."

The number of organic compounds measured by GC (59) is provided on p5 l94, species measured by PTR-MS are mentioned on p5 l98-103.

**Comment:** Authors report only acetaldehyde and sum of MVK and MACR as species measured by PTR-MS only, other species measured by PTR-MS being also measured by GC system (isoprene, benzene, toluene, styrene, C8-aromatics, C9-aromatics) (see P5, line 98-103). If so, PTR-MS measurements seem under-exploited (see de Gouw and Warneke, Mass Spectrom. Rev., 26, $223 - 257$, 2007, for a review). Did you really measure so few compounds with PTR-MS during the campaign? If more compounds were measured by PTR-MS, it should be clarified in the section 2 of the manuscript.

**Response:** Indeed, only these compounds were measured during this campaign due to the lack of calibration for other compounds that can be additionally detected by this kind of instrument.

**Comment:** Furthermore, no description of NO2 or photolysis frequency measurements is made in the section 2, while these measurements are used for estimation

of calculated OH reactivity and OH production rate, respectively.

**Response:** We rephrase the text on p4 l85 to explain how NO2 was measured: "Nitrogen oxides (NO and $NO_2$) were also detected by several instruments applying chemiluminescence technique (Thermo Electron model 42i $NO$-$NO_2$-$NO_x$ analyzer and Eco Physics model TR 780) that were equipped with a photolytic converter." We add information about photolysis frequency measurements on p5 l103 in addition to specifications given in Table 1: "Photolysis frequencies were calculated from the spectral actinic photon flux density measured by a spectrometer that was calibrated against absolute irradiance standards (Bohn et al., 2008)."

**Comment:** P4, line 85: "Nitrogen monoxide was also detected by several instruments". Please indicate how many instruments measured NO as well as their model and brand.

**Response:** We add more information about instrument models and brands for NOx (see above).

**Comment:** P4, line 86: "Measurements from one of the instruments". Please detail which instrument it is.

**Response:** We rephrase this statement: "Measurements from of the Thermo Electron instruments appeared to be more precise and are taken here (see Tan et al., 2016 for details)."

**Comment:** P5, line 89-90:"Nitrous acid (HONO) concentrations were simultaneously measured by several instruments applying different measurement techniques". Please specify which instruments were used to measure HONO (brand, model, technique).

**Response:** We add on p5 l90: "Custom-built instruments from FZJ (Li et al., 2014) and from PKU (Liu et al, 2016) utilized long-path absorption photometry (LOPAP). In addition, three custom-built instruments applied cavity enhanced absorption spectroscopy (CEAS) for the detection of HONO. They were operated by the US National Oceanic and Atmospheric Administration (NOAA) (Min et al., 2016), by the Anhui Institute of

Optics and Fine Mechanics (AIOFM), and by the University of Shanghai for Science and Technology (USST). A gas and aerosol collector (GAC), which is based on the wet denuder/ion chromatography technique, could also detect HONO (Dong et al., 2012). Only measurements by the two LOPAP instruments and the CEAS by NOAA resulted in good data coverage."

**Comment:** P5, line 90-91: "The agreement between instruments was diverse". Please develop this statement.

**Response:** We add: "Differences were often less than 30 %, but could be as high as a factor of two for certain periods (several hours). The reason for the disagreement during these times is not clear."

**Comment:** While the authors observed an imbalance between total OH production and OH destruction rates, especially in the late afternoon and at night when NO concentrations are low, only few hypotheses, from literature, are given to explain it. It would be interesting to investigate further this observed discrepancy to identify potential unaccounted OH sources in POH calculations in Wangdu.

**Response:** We agree that further insights would be good to have. However, measurements in this campaign did not give hints about the nature of a possible unknown OH source. Moreover, the difference is hardly significant as discussed on p14 l415.

**Comment:** P13, line 399-400: "Ozonolysis of alkenes species made only a minor contribution to the OH production at all time". This is not necessarily expected in anthropogenic dominated environments where these reactions can represent an important fraction of OH production in the late afternoon and at night (e.g. Ren et al., Atmos. Environ., 37, $3639 - 3651$, 2003; Kanaya et al., J. Geophys. Res., 112, 2007; Dusanter et al., ACP, 9, $6655 - 6675$, 2009), precisely the time period when the largest imbalance between POH and DOH is observed. How many and which alkenes were measured? Is it possible that an underestimation of the contribution of ozonolysis of alkenes in OH production rate, due to unmeasured alkenes, is, at least partly, responsible for the

discrepancy observed between POH and DOH in the late afternoon and at night?

**Response:** GC measurements provided C2-C6 alkene concentrations (see Table 1). Ozonolysis reactions from unmeasured alkene cannot be excluded to contribute to missing OH production, specifically monoterpene species were not measured. However, the good agreement between measured and calculated OH reactivity does not hint that a large fraction of alkene species were not measured. We add on p13 l400: "Only C2 to C6 alkene species were measured, so that ozonolysis reactions of undetected alkene species (potentially monoterpenes) could have additionally contributed to the OH production. However, the good agreement between measured and calculated OH reactivity does not hint that a large fraction of alkene species are missed."

Previous measurements indeed give partly higher contributions from ozonolysis reactions to the OH production. However, NMHC concentrations in Tokyo and Mexico City (Kanaya et al., Dusanter et al.) were much higher compared to concentrations measured in this campaign. The total OH production rate from ozonolysis reactions in New York City was not high. We add on p15 l460: "The contribution of alkene ozonolysis to the OH production in other campaigns in urban environments were partly significantly higher (Kanaya et al., 2007; Dusanter et al., 2009; Elshorbany et al, 2009) compared to the Wangdu site due to higher alkene concentrations."

**Comment:** Figure 2: Large discrepancies are observed between DOH and POH on 10 and 15 June. Maybe these days could be studied in more details to investigate potential missing OH source. At least, the large imbalance between OH production and destruction rates observed these two days could be discussed in the manuscript.

**Response:** We also hoped that the extended set of measurements during this campaign would allow identifying reasons for discrepancies like observed on these days. However, no hint was found in the measurements.

**Comment:** P9, line 246: Please indicate how many and which species are considered in the estimation of calculated OH reactivity? What are the reaction rate constants

used? All these information could be given, for example, in a table in supplementary material. These information are important to estimate the representativeness of missing OH reactivity.

**Response:** The conclusion of this campaign is that there is overall only little missing reactivity during the first part of the campaign which could be related to local emissions. All species mentioned in Table 1 were included in the calculation. Reaction rate constants were taken from IUPAC or from reaction rate constants derived from structure-activity relationship (SAR) as stated in the Master Chemical Model. We do not think that an explicit list of all rate constants is necessary, but we give more details on p9 l255: "The calculated reactivities were determined from measured CO, $CH_4$, $C_2$ to $C_{11}$ alkanes, $C_2$ to $C_6$ alkenes, $C_6$ to $C_{10}$ aromatics, formaldehyde, glyoxal, acetaldehyde, MVK, MACR, NO, $NO_2$, $SO_2$ (Table 1). Reaction rate constants were taken from IUPAC recommendations (IUPAC) or structure-activity relationship (SAR) as stated in the Master Chemical Model (http://mcm.leeds.ac.uk/MCM/)."

**Comment:** P9, line 247-249: "Because of the similarity of diurnal profiles of observations during the first and the second part of the campaign, measured kOH and calculated reactivity from major contributors are shown as median diurnal profiles with percentiles in Fig. 5". I do not understand this statement since median diurnal profiles of the first and the second part of the campaign are presented separately in figure 5. Please clarify.

**Response:** We rephrase the sentence: "During each of the two parts of the campaign (before and after 19 June), diurnal profiles of observations appear to be similar. Therefore measured $k_{OH}$ and calculated reactivity from major contributors are shown as median diurnal profiles with percentiles for each period in Fig. 5."

**Comment:** P11, line 310-313: "Largest differences of 5 to 6 s$^{-1}$ (approximately 20 %) occurred during nighttime and early morning during the first two weeks of the campaign, when also nitrogen oxide concentrations were highest. This could hint that

unmeasured OH reactants were emitted concurrently with nitrogen oxides in combustion processes". Can you correlate the missing reactivity to several source tracers (e.g. NOx, Acetonitrile etc...) trying to identify the nature of the OH reactants responsible for missing reactivity, especially during the first part of the campaign?

**Response:** We did this kind of correlations, when we analyzed our data. Unfortunately, no further measured trace gas could be identified, which correlates with missing reactivity that would give additional information about the nature of missing reactivity. We add on p11 l313: "Therefore, there is no clear further hint about the nature of missing reactivity during this period. Emissions of organic compounds from biomass burning may have not been detected during the first part of the campaign. During nighttime also near-by sources for OH reactants as indicated by the short duration of high reactivity could have contributed to the missing reactivity."

**Comment:** P11, line 322: "the photochemical age of air masses was short". Can you make an estimate of photochemical age of air masses during the campaign to support this statement?

**Response:** The photochemical age of air masses is not easily determined from measurements (like from concentration ratios) during this campaign. This is most likely due to the heterogeneity of emissions that contributed to the air masses that were encountered at measurement site and led to a mixture of air masses with different photochemical age. Therefore, we see this statement only as a possible explanation.

**Comment:** Figure 2: High concentrations of isoprene (up to 4 ppbv for example on 26 and 28 June) are sometimes observed after sunset. What are the sources of isoprene at night? Could it be due to interferences? These high concentrations lead, in particular, to large OH reactivity from isoprene in the late afternoon (after 18:00) and at night especially during the second period of the campaign (see Figure 5).

**Response:** There is no indication that isoprene GC measurements were impacted by interferences. Transport of residual isoprene that was not oxidized during daytime

could be the reason for elevated concentrations in the early evening. We add on p10 l302: "Isoprene also contributed to the reactivity in the early evening most likely because isoprene that was emitted during daytime was only partly oxidized by OH before sunset."

**Comment:** Figure 8: Dark grey area should also be defined in the legend.

**Response:** Because this grey area is only the difference between OH production and destruction rate and does not originate from calculations of a OH production rate using measurements, we think that this is qualitatively different from the other colored areas. It is only meant to guide the eye, but does not add in the same way to the other colored areas. The meaning is already explained in the caption, so that we do not think that changes are needed.

Other minor comments are corrected as suggested by the reviewer.

---

## Author Comment (AC2) · 24 Nov 2016

We thank the reviewer for the helpful comments.

**Comment:** I would suggest to add : - more connections with the conclusions from the accompanying paper (Tan, 2016) in the present paper, particularly because the modeled OH reactivity is presented in (Tan, 2016) as well as sensitivity run constrained on the basis of OH reactivity measurements, - a more detailed discussion on the missing reactivity, - gathering the discussion on the comparison with previous campaigns to avoid repetition and to help the reader to better see the new understanding brought by this campaign and in particular the reactivity measurements.

[Figure]

**Response:** We agree that there should be a close connection to the paper by Tan et al.. We try to improve this point as can also be seen in the revision based on comments from all reviewers.

The modelled OH reactivity presented in Tan et al. does not differ significantly from calculations presented in this paper, because only modelled PAN and aldehyde species added to the OH reactivity. Therefore, there is no additional conclusion from that analysis for the kOH. We make a statement on p12 l320: "Therefore, concentrations of oxygenated organic compounds that are produced by model calculations but that were not detected were constrained to zero in calculations presented in our accompanying paper by Tan et al. (2016), in order to ensure that modelled OH reactivity is consistent with measurements."

We also make better use of results from model calculations derived by Tan et al by adding on p13 l408: "The result of the budget analysis is consistent with the finding by Tan et al. (2016) that model calculations underpredict OH by up to a factor of two at NO mixing ratios of less than 0.3 ppbv, but simulate HO2 and kOH correctly under these conditions at the Wangdu site. The good description of HO2 and kOH means that the major known OH source (the reaction of HO2 and NO) and the total OH loss rate are well represented by the model. Further model tests suggest a missing process that recycles OH from RO2 and HO2 by an unknown agent that behaves like 0.1 ppbv NO (Tan et al., 2016). Other trace gases measured at Wangdu give no hint to the nature of the missing source in the OH budget analysis or in the model results."

More discussion on missing reactivity are added. Please refer to details to the answers of the comments by reviewer #1.

We have the feeling that the discussion on the comparison with previous field campaigns is appropriate and would like to keep it as it is.

**Comment:** L29 : It is mentioned in the introduction that "the effort to improve our knowledge of radical chemistry in Chinese megacity areas was continued by a comprehensive field campaign at a location close to the city Wangdu" please resume the conclusion of the accompanying paper (Tan, 2016) and put in the conclusion of the article the key results from this campaign which contribute to this improvement. What type of environment should be studied in future campaigns to bring complementary information as in the present study the low reactivity and potential OH interferences seem to prevent to draw clear conclusion concerning the OH budget?

**Response:** We extend the introduction on p3 l33: "Compared to our previous field campaigns in China 2006 (Hofzumahaus et al., 2009), the quality and number of measurements have been improved. A large number of instruments measured a variety of different trace gases, part of which were simultaneously detected by several instruments. Specifically, measurements of organic oxygenated compounds such as formaldehyde and acetaldehyde were achieved, which was not the case in previous campaigns. Radical measurements were improved by performing additional tests of potential interferences in the detection of OH and a modified the detection scheme for $HO_2$ that avoids interference from $RO_2$ was applied (Fuchs et al., 2011)."

As stated in the paper of Tan et al. further improvement of the tests for potential interferences in the OH detection will be done in future campaigns to avoid additional uncertainty. Environments with a larger fraction of biogenic reactants could be of interest to complement results of this campaign. We add at the end of the conclusion: "For future field work, comprehensive studies like this campaign in photochemically active environments where larger contributions from biogenic reactants can be expected in addition to anthropogenic emissions may help to solve the still open questions of imbalances in the OH production and destruction and measured and calculated OH reactivity that have been observed in other campaigns."

**Comment:** It is confusing to see in the introduction that the site is described as "close to the city Wangdu", described at rural in the title and that the campaign location is in a botanic garden close to the "small" town Wangdu. Please clarify how and why this site has been chosen and how it is classified. Linked comment: how the comparison with

other campaigns has been chosen? L350- 365, related to anthropogenic emissions but without the comparisons with reactivity measurements done in Paris or Mexico, whereas these campaigns are discussed later (l453-460) concerning HONO. I would propose to gather the comparison paragraphs.

**Response:** We state on p4 l69: "The site was chosen, because it was not directly influenced by strong close-by anthropogenic emissions or the direct outflow of a big city. However, it was expected to observe regionally transported pollution in the North China Plain."

As shown in the discussion of results, the measurement site was mainly influenced by emission of anthropogenic sources. Therefore, the comparison to results from other campaigns is done for campaigns in urban environments during summertime, which provided information about kOH and the OH budget from measurements (see also the review by Yang et al. Atmos Environ 134, 147-161, 2016). We add a statement regarding the measurements done in Mexico on p12 l355: "Significantly higher morning values of $130\,\mathrm{s}^{-1}$ were observed in Mexico City 2003 (Shirley et al. 2006)." The section p15 l453-460 specifically discusses the impact of HONO photolysis for the OH budget, so that we also included measurements in Paris, although they were performed during wintertime. We rephrase the statement on p15 l457: "These campaigns took place in or very close to very large cities the one in Paris during wintertime) and NO concentrations were often exceptionally high, so that HONO formation was favored." For the same reason we do not think that it should be moved to another position in the manuscript.

**Comment:** L73: Particle measurements are also available. Could it be commented?

**Response:** Particle measurements were done, but a detailed discussion is out of the scope of the paper. We add PM2.5 measurements in Fig. 2 and add text on p9 l234: "Typical daytime maximum PM2.5 concentrations ranged between 30 and 90 $\mu$ g/m$^3$ but were as high as 300 $\mu$ g/m$^3$ on one day due to the local biomass burning. No clear connection between OH reactivity and aerosol number concentration was observed.

Although a sharp drop in PM2.5 was observed on 19 June when also OH reactivity dropped, PM2.5 increased again to higher values till the end of the campaign."

**Comment:** L156: how the delay to start the fit is defined (which level of deviation is considered to discard the points?)

**Response:** We add on p6 l157: "The fit is started, if the count rate has decreased to the 90 % level of the maximum count rate."

**Comment:** L157: The reason for the deviation is attributed to the non homogeneity of the OH distribution: could this be clarified? Is it due to the heterogeneous distribution of the laser energy?

**Response:** We add on p6 l157: "The likely reason is that the spatial OH distribution is not perfectly homogeneous near the inlet nozzle of the OH detection cell right after the laser pulse due to inhomogeneities in the laser power across the laser beam."

**Comment:** L165: which species considered as contaminations have been identified in the gas cylinder?

**Response:** The major contamination that was measured by the GC system was toluene (50%) and smaller contributions from mainly butene and acetaldehyde, but also a larger number of other reactants. Concentrations were rather small and therefore the measurement of this contamination has some uncertainty. As stated in the text we therefore consider the correction of data for this contamination as additional uncertainty in our measurements.

**Comment:** L184: why the assumption of a bi-exponential fit would better describe the conditions with recycling?

**Response:** A bi-exponential behavior of the OH is expected from reaction kinetics. However, the attribution of the faster decay time to the OH reactivity is only approximately valid within certain limits of chemical conditions (see also Lou et al, ACP 2009). A detailed description is beyond the scope of this paper and also not relevant, because

no bi-exponential fitting was applied for measurements in this campaign. We add on p7 l184: "This can be derived from reaction kinetics. The faster decay time represents approximately the OH reactivity for certain chemical conditions."

**Comment:** L189: which criterion is used to decide that the measurements appear as single exponential decays?

**Response:** Deviation from a single exponential decay would be clearly seen in the residuum of the fit which is calculated for each individual fit. We add on p7 l188: "... no bi-exponential behavior was observed that would have been seen in the residuum of the fit."

**Comment:** L206 : would be useful to show other correlations with individual species (provide also more details in the repartition of the reactivity for specific periods (L310)

**Response:** We tried correlation with other species, but no other OH reactant could be identified that would give additional insights to differences between the two parts of the campaign (see also responses to comments from reviewer #1).

**Comment:** L214/Figure 4: difficult to identify the different trajectories

**Response:** We tried to visualize trajectories best. For the interpretation of the result, it is not necessary to identify every single trajectory. An alternative presentation would result a larger set of figures, which would not give additional information.

**Comment:** L232: other species correlate with acetonitrile (could provide the profile in Figure 2)? New peaks (GC, PTR-MS) have been observed during the biomass activities?

**Response:** We add acetonitrile measurements in Fig. 2. Unfortunately, no clear correlation of acetonitrile measurements with other trace gases could be identified most likely because concentrations of other species were influenced by other sources or chemical transformation.

**Comment:** L264 and 302: influence of products due to oxidation by nitrate radical could be discussed, parallel with particle profiles could be discussed.

**Response:** We think that a statement of nighttime oxidation products is best placed on p11 l314: "In addition, undetected products from the oxidation by the nitrate radical could have been part of missing reactivity in the night."

See our response above regarding particle measurements.

**Comment:** L280-291: some repetitions with 3.1

**Response:** We move text from p9 l256-261 and merge this with text on p10 l280-291.

**Comment:** L297: "It is most prominently seen in median alkene and alkane concentrations". Where?

**Response:** We rephrase this statement: "It is most prominently seen in median alkene and alkane concentrations during nighttime (Fig. 5)."

**Comment:** L322: possible to use ratio of species with different rate constants with OH to estimate this photochemical age?

**Response:** Please refer to our answer to the same comment by reviewer #1.

**Comment:** L350-365: see comment above. What can be concluded from these comparisons?

**Response:** The conclusion from this comparison is that the reactivity at the measurement site is typical for an environment that is influenced by anthropogenic activities. We add on p350 l350: "The OH reactivities measured at the Wangdu site in the North China Plain show diurnal profiles that are comparable to those reported for other polluted environments all over the world (see review by Yang et al. 2016).".

**Comment:** L382: What is missing. How it has been concluded that "results do not change significantly, whether the first part is included or not"? Please detail.

**Response:** We add the requested information on p13 l382: "Unfortunately, the data coverage of simultaneous measurements before 20 June (mostly due to missing radical measurements) is not sufficient...".

Results are mainly based on the analysis of median diurnal profiles, which were also calculated for only the second part of the campaign. None of the points we discuss in the paper changes, if all data are included or only the second part is taken into account. We rephrase this sentence: "However, results do not change significantly, whether the first part is included in the median diurnal profiles that are discussed below or not."

**Comment:** L419: possible to show the results with the bias subtracted?

**Response:** We would like to emphasize that there is no clear evidence for a bias in the OH measurements as discussed in detail in the paper by Tan et al. 2016. A possible bias is within the uncertainty of tests for potential interferences in the OH measurements that we performed during the campaign. Therefore, there is no reason to subtract a bias.

**Comment:** L432: Please detail.

**Response:** A detailed discussion of a potential interferences in OH measurements and the method how tests were performed during the campaign is given in our accompanying paper by Tan et al. 2016. Consequences for the accuracy of the analysis of the OH budget are clearly stated in this manuscript.

**Comment:** L439: why introducing the term turnover rates there? Useful? Check the consistency of the terms used in the Figure 8.

**Response:** We change the term to "production and destruction rates" at this position and also change the label in Fig. 8 accordingly.

**Comment:** L475: but there are periods with significant missing reactivity. It could be mentioned in the conclusion.

**Response:** We add on p15 l468: "Highest missing reactivity of the median diurnal profile (approximately 25%) was observed during nighttime of the first part of the campaign, which could have been related to nearby emissions or undetected oxidation products.".

---

## Author Comment (AC3) · 24 Nov 2016

We thank the reviewer for the helpful comments.

**Comment:** The comparisons between measured and calculated OH reactivity are presented only as time series, the manuscript would benefit from a scatterplot showing the calculated OH reactivity against the measurements, enabling a more direct comparison.

**Response:** We add a scatter plot and add text on p11 l309: "The good agreement between measured and calculated OH reactivity is also demonstrated by the high linear correlation coefficient ($R^2 = 0.77$ for the entire data set and both subsets of data)

between both values (Fig. new). For the second part of the campaign a linear regression analysis yields a slope of 0.96 with a negligible intercept of $-0.33\,\mathrm{s}^{-1}$. As already discussed missing reactivity was higher during the first part of the campaign, so that a regression analysis yields a higher slope of 1.7 with an intercept of $-4.2\mathrm{s}^{-1}$. The larger intercept is due to a slightly non-linear relationship between measured and calculated reactivity for this subset of data."

**Comment:** The manuscript reports a limited role for oxidation intermediates. Model results reported for OH concentrations in the manuscript by Tan et al. could be used to quantify the contributions of unmeasured oxidation intermediates to the total OH reactivity, but are only briefly discussed.

**Response:** Please refer to our answers to reviewer #2.

**Comment:** Abstract: Mention the technique used to make the OH reactivity measurements.

**Response:** We add on p1 l8: "Total OH reactivity was measured by a laser flash photolysis - laser induced fluorescence instrument (LP-LIF)."

**Comment:** Lines 7-8: Quantify "good correlation" and "high contribution".

**Response:** We rephrase p1 l7-8: "...by a good correlation between measured OH reactivity and carbon monoxide (linear correlation coefficient $R^2 = 0.33$), and (2) by a high contribution of nitrogen oxide species to the OH reactivity (up to 30 % in the morning)."

**Comment:** Line 63-69: Can you comment on the prevailing wind direction?

**Response:** A discussion of the origin of air masses is given by back trajectory calculations presented in Fig. 4 and discussed on p8 l209-223.

**Comment:** Line 84: quantify the agreement (and elsewhere, e.g. line 102).

**Response:** We will rephrase the statement on p4 l84: "...agreed well within their accuracies..." and on p4 l102: "...which during daytime in general agreed with measurements by GC within 30 to 50%...". See also answers to reviewer #1 for further modifications concerning this comment.

**Comment:** Line 85: How many instruments? What was the standard deviation of the average? How did one of the instruments "appear to be more precise"? How does the uncertainty in the NO measurements impact the analysis of OH reactivity?

**Response:** Please refer also to our answer to a similar comment from reviewer #1. "More precise" means that the reproducibility of calibration was better for one of the instruments. The impact of the additional uncertainty of the NO measurements on the analysis is taken into account in the estimate of uncertainties given on p11 l304-309 and shown in Fig. 8.

**Comment:** Line 93: How small is "rather small"? Quantify the impact.

**Response:** We rephrase the statement: "The choice of the HONO data set has a rather small impact on calculated OH reactivity, as well as on the calculated total OH production rate which was dominated by OH recycling from $HO_2$ during daytime (see below)." Please refer also to our answers to comments from reviewer #1.

**Comment:** Line 101: "part of the same species" − please rephrase to clarify the meaning.

**Response:** We rephrase this sentence: "Some of the species or family species were simultaneously detected by the GC system and the PTR-MS..."

**Comment:** Line 113: Please clarify that it is the previously reported Zeppelin instrument that is being used in this campaign.

**Response:** We will add on p6 l157: " This instrument was deployed in this campaign."

**Comment:** Line 121: Does the length of the sampling line affect the measurements?

**Response:** A sampling line of similar length and the same coating has been used

for the measurement of OH reactivity in our simulation chamber SAPHIR for a variety of different chemical conditions. So far, no unexpected difference between measured and calculated has been observed that would hint to an impact of the inlet line on measurements. Therefore, we are confident that there are no significant effects on measurements from the sampling line. We will add on p5 l122: "Such sampling line has been used for OH reactivity measurements in the Jülich atmosphere simulation chamber SAPHIR for many years without notable effects on measurements."

**Comment:** Line 126: How does the change in temperature affect the measured OH reactivity? What is the difference from the external ambient temperature?

**Response:** We will add on p6 l126: "Ambient temperature was higher with up to 38 °C for some periods during the campaign. Differences in temperature and pressure potentially effect the measured reactivity due to changes of the reactant concentrations and of reaction rate constants. Measured reactivities were corrected for changes in the reactant concentration calculated from measured ambient and flow-tube temperature and pressure values (corrections were less than 2 %). Sensitivity studies taking either ambient temperature or flow-tube temperature for the calculation of OH reactivity from measured OH reactant concentrations (see below) indicate that the effect of temperature differences on reaction rate constants resulted in changes in the OH reactivity of typical less than 1 % (maximum values 4 %) for conditions of this campaign."

**Comment:** Line 150: Clarify what you mean by "sufficiently precise". How does the summing of decay curves affect the reported values? Does it make any difference to simply average 60 decays before fitting as opposed to summing ten decays and then averaging six summed decays?

**Response:** Typical maximum counts for a single decay curve that was evaluated were 60 to 100 counts during this campaign. From counting statistics this gives an error of 10%. In this case, the fit result does not change significantly, if more traces are added. We will add the typical maximum counts on p6 l150. Changes in the decay

time within the time of summing/avering are small enough that results are not affected by this procedure other than that the precision is improved. Summing up and averaging has the advantage that realistic errors can be used for the fit procedure as stated in the text. Realistic errors can give more accurate results, if the noise of the decay curve is different from the assumption of pure shot noise. Both approaches have been regularly used by us in various campaigns. Results agree within their uncertainties in most cases.

**Comment:** Line 165: What is the uncertainty/precision of the measurements?

**Response:** Uncertainties and precision of measurements are specified in Table 1. The measurement of contamination in the zero air was done with the same GC instrument that also performed measurements during the campaign.

**Comment:** Line 187: What was the mean/standard deviation/median NO?

**Response:** Please refer to Fig. 7, which gives median NO concentrations and 25 and 75 percentiles.

**Comment:** Line 203: Quantify the correlation.

**Response:** See our answer to the comment above. We will rephrase the sentence by replacing "correlated with" by "related to".

**Comment:** Lines 215-234: Much of this is poorly phrased and difficult to follow (particularly lines 224-228), please consider re-writing.

**Response:** We rephrase this paragraph.

**Comment:** Line 308: Re-iterate the source of the uncertainties and why they have been separated into two terms.

**Response:** The 10% accuracy of the OH reactivity measurements originates from consideration described in Lou et al 2009. It is based on the assumption that the accuracy is limited by the uncertainty in the reaction rate constants of CO, because

the accuracy of OH reactivity measurements were tested by measuring the reactivity from known CO concentrations. The additional uncertainty of $+0.7s^{-1}$ is campaign-specific due to the uncertainty of the zero decay (see answers above). Because the first uncertainty is a relative value and the second one is an absolute value they are stated separately.

**Comment:** Line 315: Quantify the "exceptionally good agreement".

**Response:** Please refer to the answer to the first comment.

**Comment:** Line 364: The measured OH reactivity in London contained significant contributions from model-generated intermediates.

**Response:** We correct this error.

**Comment:** Line 402 and following paragraph: Quantify "nearly balanced", "slightly larger" and "hardly significant".

**Response:** We quantify "nearly balanced" by adding on p13 l402: "The OH destruction rate is on average only 20 % higher than the sum of OH production during daytime." and cancel the statement containing "slightly larger". "hardly significant" is already quantified as "hardly significant with respect to 405 the experimental accuracies (Fig. 8)".

**Comment:** Line 411: Quantify "much larger and highly significant".

**Response:** We add "up to a factor of four" on p14 l411.

**Comment:** Line 422: What about the possibility of OH regeneration through peroxy radical reactions with the nitrate radical?

**Response:** Model calculations of OH concentrations discussed in our accompanying paper by Tan et al. 2016 indicates that NO3 chemistry did not significantly contribute to nighttime OH production.

**Comment:** Line 445: Quantify or define the level of significance in the term "clearly above the level of significance".

**Response:** We add on p14 l445 "...with respect to the measurement uncertainties".

Other minor comments are corrected as suggested by the reviewer.
* * *

---

## Author Comment (AC4) · 24 Nov 2016

We thank the reviewer for the helpful comments.

**Comment:** $305 - 310$: Most of uncertainty analysis that I have encountered uses at least 2 sigma uncertainty rather than 1 sigma. If you use 2 sigmas probably the calculated OH reactivity would be able to account measured OH reactivity.

**Response:** We fully agree with this statement. We rephrase the statement on p11 l305: "Even during times when measured reactivity was higher than calculations from OH reactants, the gap is within the combined $2\sigma$ uncertainties:"

**Comment:** Figure 7 and Figure 8: It has been highly controversial about the night-

time OH. As observed, the lifetime of OH is much shorter during the nighttime so it is rather surprising to see observed nighttime OH such a low OH production during the night. There is an obvious attempt to account the observation such as ozonolysis of terpenoids and dissociation of potential contributions from PANs but could not provide a quantitative assessment since there is no observation data. However, as it is so important issue, I think at least the authors should attempt to assess what kind of PANs or terpenoids levels you would need to account the night time OH. Otherwise, as the discrepancy between OH production rates and OH destruction rates are appeared almost identical except in the early morning, some may conclude that the discrepancy may be simply accounted by an instrument artifacts as described in the manuscript.

**Response:** There is no direct way to estimate PAN concentrations, because the impact on OH is indirect. PAN has to be transported and decomposes then to HO2/RO2, if the PAN that is transported disturbs the thermal equilibrium. An estimate would require a 1D model calculation that is beyond the scope of this paper. In the paper by Lu et al. 2014 it was shown that the impact of PAN decomposition alone made only a small HOx source of less than 0.01ppbv/h, which would not be sufficient to balance the OH loss rate. In order to balance the OH production rate by an ozonolysis reaction, the alkene concentration can be calculated. We add on p14 l431: "In order to balance the calculated OH destruction rate during nighttime, a rather large concentration of an alkene would be required. Assuming an ozone concentration of 30ppbv, a reaction rate constant for the ozonolysis reaction of $1.8 \times 10^{-15}$ cm$^3$s$^{-1}$ for $\delta$-terpine and an OH yield of one (Atkinson and Arey 2003), the concentration would need to be around 600pptv."

**Comment:** (minor comment) Recently, Kim et al (2016) reported observed OH reactivity in the Seoul Metropolitan Area. The addition of this reference to the comparison could be useful.

**Response:** We include this reference on p12.

---

## Author Response (AR2)

**Response to the comments of the editor:**

**Comment:** The discussion in the last paragraph of pg. 12 is misleading: "As already discussed, missing reactivity was higher during the first part of the campaign, so that a regression analysis yields a higher slope of 1.7 with an intercept of -4.2s" A negative intercept is not physically reasonable. I suggest that the regression be redone with the intercept forced to zero. The slope would then give a more realistic measure of the missing reactivity. If you wish, both regression analyses can be discussed, or simply the discuss the regression with the slope forced to zero.

**Response:** We change the fit procedure to force the line to zero and cancel the discussion of an intercept.

**Comment:** The sentence on lines 402-404 is not clear; please rewrite.

**Response:** We rephrased the sentence: "In addition, unmeasured oxidation products may still have contributed to the OH reactivity within the combined uncertainties of OH reactivity measurements and calculations from OH reactant measurements.